**Review article**

# Metabolomic signatures associated with fetal growth restriction and small for gestational age: a systematic review

Agustin Conde-Agudelo[1] ✉, Jose Villar [1,2] ✉, Milagros Risso[3], Aris T. Papageorghiou [1,2], Lee D. Roberts [4] & Stephen H. Kennedy [1,2]

The pathways involved in the pathophysiology of fetal growth restriction (FGR) and small for gestational age (SGA) are incompletely understood. We conduct a systematic review to identify metabolomic signatures in maternal and newborn tissues and body fluids samples associated with FGR/SGA. Here, we report that 825 non-duplicated metabolites were significantly altered across the 48 included studies using 10 different human biological samples, of which only 56 (17 amino acids, 12 acylcarnitines, 11 glycerophosphocholines, six fatty acids, two hydroxy acids, and eight other metabolites) were significantly and consistently up- or down-regulated in more than one study. Three amino acid metabolism-related pathways and one related with lipid metabolism are significantly associated with FGR and/or SGA: biosynthesis of unsaturated fatty acids in umbilical cord blood, and phenylalanine, tyrosine and tryptophan biosynthesis, valine, leucine and isoleucine biosynthesis, and phenylalanine metabolism in newborn dried blood spot. Significantly enriched metabolic pathways were not identified in the remaining biological samples. Whether these metabolites are in the causal pathways or are biomarkers of fetal nutritional deficiency needs to be explored in large, well-phenotyped cohorts.

The phenotypic term fetal growth restriction (FGR) is used to describe a highly heterogeneous syndrome characterised by the fetus' failure to achieve its genetic growth potential compared to international growth standards[1–3]. The term small for gestational age (SGA) is used to describe an infant born with a birthweight less than the 10th centile for gestational age and sex. Using such standards[1–3] to avoid the bias associated with population-specific charts[4], it has been estimated that 23.4 million newborns (17.4% of all liveborn babies worldwide) in 2020 were SGA[5]. Nearly a quarter (22.4%) of the 2.4 million neonatal deaths worldwide were attributable to preterm (<37 weeks' gestation) or term (≥37 weeks' gestation) SGA[5], and 21.2% of stillbirths at ≥22 weeks' gestation were SGA[6].

Both growth-restricted and SGA fetuses are at higher risk of perinatal morbidity and mortality compared with non-growth-restricted and/or appropriate for gestational age (AGA) fetuses[7–10]. In addition, surviving growth-restricted and SGA infants have an increased risk for death, stunting, wasting, neurodevelopmental impairment during childhood, reduced intelligence quotient and cognitive performance, autism spectrum disorders, depression, and chronic diseases in adulthood[10–18].

A wide range of analytical methods have been employed to screen for FGR; however, none of the biomarkers proposed to date are sufficiently accurate for screening, prevention, treatment development or routine clinical practice[19,20]. Metabolomics, despite its limited use in

[1]Oxford Maternal & Perinatal Health Institute, Green Templeton College, University of Oxford, Oxford, UK. [2]Nuffield Department of Women's & Reproductive Health, University of Oxford, Oxford, UK. [3]Hospital Universitario General de Villalba, Madrid, Spain. [4]Leeds Institute of Cardiovascular and Metabolic Medicine, University of Leeds, Leeds, UK. ✉e-mail: condeagu@hotmail.com; jose.villar@wrh.ox.ac.uk

clinical practice, may be a more suitable methodology[20–22] having accelerated understanding of metabolic diseases and detected silent phenotypes (only present in specific physiological conditions)[23], thereby establishing biomarkers that precede disease pathology[24–26].

Therefore, a critical appraisal of the existing metabolomic evidence is required to shed new light on the metabolic pathways involved in the pathophysiology of FGR/SGA[13]. So, we conducted a systematic review aiming to identify metabolomic signatures in tissues and biofluids of pregnant women, placentas, umbilical cords and newborns associated with FGR/SGA compared to the corresponding reference group.

## Results

### Selection, characteristics and risk of bias of studies

Figure 1 summarises the process of identification and selection of studies. Our search strategy identified 3134 citations. After removing duplicates and clearly ineligible records, we assessed 115 potentially eligible studies for possible inclusion, from which we excluded 67 based on parameters outlined in our methodology. Forty-eight studies[27–74], which included a total of 4228 women and 820,271 newborns, met the inclusion criteria, reflecting the paucity of mother-offspring dyad studies.

The main characteristics of included studies are presented in Table 1. Nineteen studies were conducted in Europe[28,30–32,34–36,40,42,43,45,46,49,54,56,58,59,64,68,69], 14 in Asia[37,38,48,51,52,55,60,61,63,66,68,70,72,74], eight in the United States[33,44,47,50,57,62,71,73], two in Australasia[41,65], four in both Europe and Australasia[27,29,53,67], and one in Egypt[39]. There were 37 case-control studies[27–32,34–41,43,44,46–54,56–60,66,68–72,74], eight cross-sectional studies[33,42,45,55,61–63,67], two cohort studies[64,73] and one case-cohort study[65]. The sample sizes ranged from 17[27] to 878[64] women (median, 77) and 14[29] to 736,435[62] newborns (median, 80). The number of cases ranged from 9[27] to 147,287[62] and the corresponding number of controls ranged from 8[27] to 589,148[62]. Thirty-five (73%) studies had <50 FGR or SGA cases.

Metabolomics were assessed in the following biological samples: maternal plasma or serum (19 studies[29,31,32,40,42,43,48–50,52–54,56,64,65,70,72–74]), maternal urine (three studies[36,47,53]), maternal hair (two studies[37,41]), maternal faeces (one study[70]), amniotic fluid (one study[66]), placenta (five studies[27,57,60,71,72]), umbilical cord blood (21 studies[29–32,39,40,42–45,54,56,58,59,63,64,67,72–74]), newborn dried blood spots (seven studies[33,38,51,55,61,62,68]), newborn urine (three studies[28,35,69]), and breast milk (one study[46]). Fourteen studies[29,31,32,40,42,43,53,54,56,64,70,72–74] collected multiple biological samples (Table 1). Among the studies that assessed metabolomics in maternal samples, only 7 reported on fasting status at the time of sample collection: in six studies, the time elapsed between last food intake and sampling was at least eight hours[31,43,50,53,54,56], whereas in the remaining study[74] the samples were collected in a "fasting state".

Gestational age was determined from the woman's last menstrual period alone in one study[55], from the woman's last menstrual period and confirmed by ultrasound in the first trimester in 11 studies[32,34,36,43,47,49,50,56,60,64,68], and from the woman's last menstrual period and confirmed by ultrasound in either the first or second trimester in four studies[30,40,52,71]. The remaining 32 studies did not report on the methods used for determining gestational age. Twenty-one studies included only term infants[27,29,31,32,36,37,41–43,45,47,49,51,53–55,59,61,72,74], three included only preterm infants[28,33,69,] and 24 included both preterm and term infants[30,34,35,38–40,44,46,48,50,52,56–58,60,62,64–68,70,71,73]. None of these 24 studies reported results separately for preterm and term infants.

The case definitions included the following: birthweight for gestational age <10th customised (six studies[29,36,41,46,53,54]) or non-customised (22 studies[31–33,37–39,42,44,45,47,50,51,55,57,61–64,68,71,73,74]) centile;

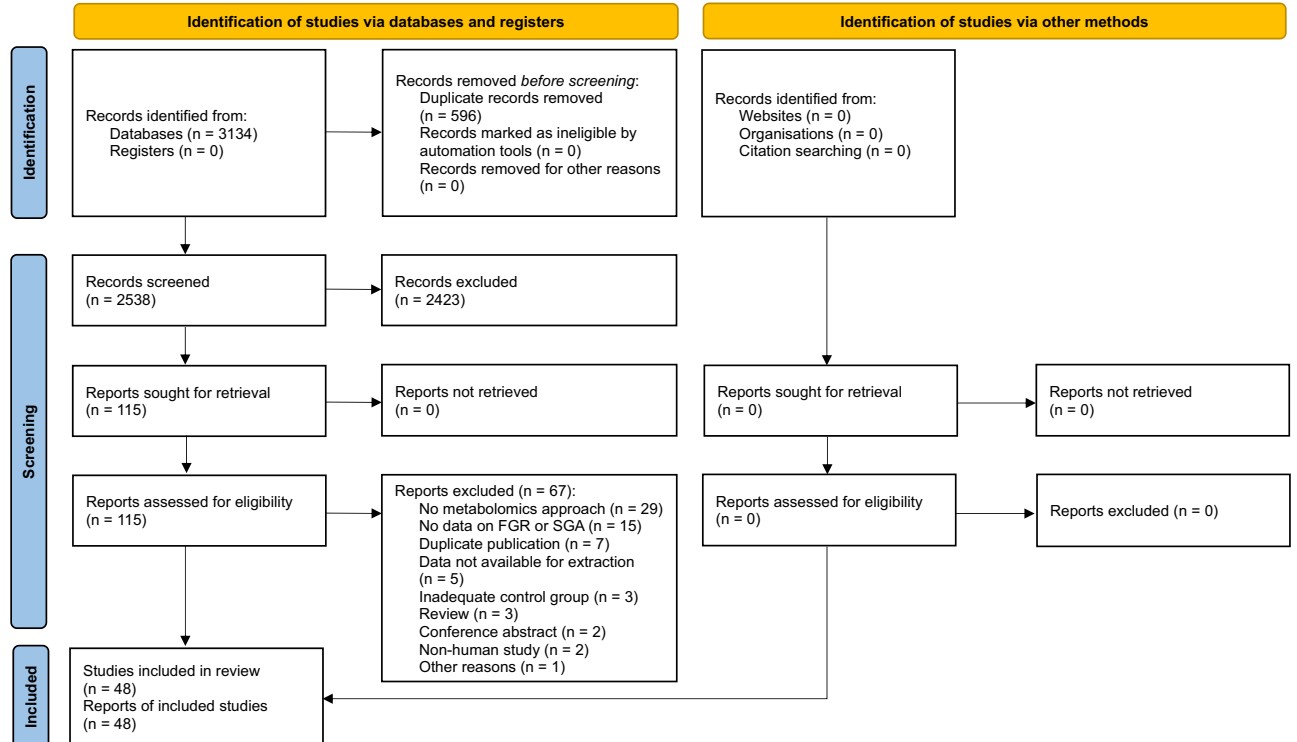

**Fig. 1 | PRISMA flow diagram.** This figure illustrates the PRISMA (Preferred Reporting Items for Systematic Reviews and Meta-Analyses) flow diagram detailing the study selection process. The diagram includes the number of records identified, screened, assessed for eligibility, and included in the systematic review. FGR fetal growth restriction; SGA small for gestational age. Source: Page MJ, McKenzie JE, Bossuyt PM, Boutron I, Hoffmann TC, Mulrow CD, et al. The PRISMA 2020 statement: an updated guideline for reporting systematic reviews. BMJ 2021;372:n71. doi: 10.1136/bmj.n71. For more information, visit: https://www.prismastatement.org/prisma-2020-flow-diagram.

**Table 1 | Main characteristics and findings of studies included in the systematic review**

| First author, year (Country) | Case definition[a] (n) | Control definition[a] (n) | Mean or median gestational age at birth | Biological sample; sampling time | Metabolomics approach | Analytical platform used | Significantly up-regulated metabolites | Significantly down-regulated metabolites |
|---|---|---|---|---|---|---|---|---|
| Horgan[27], 2010 (United Kingdom and Australia) | Women giving birth to an SGA baby (birthweight <5th customised centile) (n = 9) | Women with an "uncomplicated pregnancy" giving birth to a healthy term baby (n = 8) | Cases: 38.4 weeks Controls: 39.1 weeks | Placenta; within 20 min of delivery | Untargeted | UPLC-MS | Unclear[b] | Unclear[b] |
| Dessì[28], 2011 (Italy) | Preterm newborns with "IUGR diagnosed ultrasonographically in the prenatal period" and birthweight <10th centile (n = 26) | Preterm newborns with a "suitable weight for their gestational age at birth" (n = 30) | Cases: unreported Controls: unreported | Neonatal urine; in the first 24 h prior to feeding and about 4 days after birth | Untargeted | ¹H NMR | Myo-inositol; creatinine; sarcosine; creatine | None |
| Horgan[29], 2011 (United Kingdom and Australia) | SGA: women giving birth to a term baby with a birthweight <10th customised centile (n = 48: 40 women and 8 newborns) | Women with an "uncomplicated pregnancy" giving birth to a healthy term baby (n = 46: 40 women and 6 newborns) | Cases: 39. 6 weeks (women) and 39.0 weeks (newborns) Controls: 40.2 weeks (women) and 38.5 weeks (newborns) | Maternal plasma; at 15 weeks' gestation (n = 80) Cord blood; at birth (n = 14) | Untargeted | UPLC-MS | Maternal blood: leucyl-leucyl-norleucine; sphingosine 1-phosphate; cervonyl carnitine; 1α,25-dihydroxy-18-oxocholecalciferol; lysoPC (16:1); sphinganine 1-phosphate; 6-hydroxysphingosine; (4OH,8Z,t18:1) sphingosine; 15-methyl-15-PGD2; 15R-PGE2 methyl ester Cord blood: DG (32:0) | Maternal blood: phosphocholine; pregnanediol-3-glucuronide; 3α,20α-dihydroxy-5β-pregnane 3-glucuronide Cord blood: leucyl-leucyl-norleucine; sphingosine 1-phosphate; cervonyl carnitine; 1α,25-dihydroxy-18-oxocholecalciferol; (15Z)-tetracosenoic acid; 10,13-dimethyl-11-docosyne-10,13-diol; trans-selacholeic acid; hexacosanedioic acid; pentacosenoic acid; teasterone; typhasterol; lysoPC (18:2); ubiquinone-8; lysoPC (16:1); pregnanediol-3-glucuronide; 3α,20α-dihydroxy-5β-pregnane 3-glucuronide; 6-hydroxysphingosine; (4OH,8Z,t18:1) sphingosine; 15-methyl-15-PGD2; 15R-PGE2 met, hyl ester |
| Favretto[30], 2012 (Italy) | IUGR: EFW <10th centile for gestational age by third trimester ultrasound, and confirmed at birth (n = 22) | AGA: EFW between the 10th-90th centiles for gestational age by third trimester ultrasound, and confirmed at birth (n = 21) | Cases: 38.0 weeks Controls: 38.3 weeks | Cord blood; at birth | Untargeted | LC-HRMS | Phenylalanine; tryptophan; methionine; proline; valine; isoleucine; glutamate; dopamine; histidine; uric acid; caffeine; 5-methyl-2-undecenoic acid; leu pro; L-thyronine; hexadecanedioic acid; arginine cysteine asparagine; arginine phenylalanine arginine; tryptophan arginine; 1-hydroxyvitamin D₃ 3-D-glucopyranoside | None |
| Ivorra[31], 2012 (Spain) | SGA: term neonates with a birthweight <10th centile (n = 20) | AGA: term neonates with a birthweight between the 75th and 90th centiles (n = 30) | Cases: 37.6 weeks Controls: 39.0 weeks | Maternal plasma; 2-4 h after delivery Cord blood; at birth | Untargeted | ¹H NMR | Maternal blood: none Cord blood: citrulline; phenylalanine | Maternal blood: none Cord blood: proline; free choline; glutamine; alanine; glucose; glycogen fragments |
| Bobiński[32], 2013 (Poland) | SGA: term neonates with a birthweight <10th centile (n = 23) | AGA: term neonates with a birthweight between the 10th and 90th centiles (n = 54) | Cases: 38.0 weeks Controls: 39.2 weeks | Maternal serum; 3-5 h before birth Cord blood; at birth | Targeted | GC-MS | Maternal blood: none Cord blood: dodecanoic (lauric) acid | Maternal blood: none Cord blood: stearic acid; gamma-linolenic acid; arachidic acid; eicosatrienoic acid; arachidonic acid; Σn-6 polyunsaturated fatty acids |

**Table 1 (continued) | Main characteristics and findings of studies included in the systematic review**

| First author, year (Country) | Case definition[a] (n) | Control definition[a] (n) | Mean or median gestational age at birth | Biological sample; sampling time | Metabolomics approach | Analytical platform used | Significantly up-regulated metabolites | Significantly down-regulated metabolites |
|---|---|---|---|---|---|---|---|---|
| Ryckman[33], 2013 (United States) | SGA: preterm neonates with a birthweight <10th centile (n = 47) | AGA: preterm neonates with a birthweight between the 10th and 90th centiles (n = 374) | Cases: unreported Controls: unreported | Dried blood spots (heel prick) for newborn metabolic screening; 24–72 h after birth | Targeted | LC-MS | Alanine; C0; C2; C18:2 | Tyrosine |
| Sanz-Cortés[34], 2013 (Spain) | IUGR: neonates with a birthweight <10th centile (n = 76: early-onset IUGR [abnormal umbilical artery Doppler and delivery <35 weeks' gestation; n = 20] and late-onset IUGR [normal umbilical artery Doppler and delivery ≥35 weeks' gestation; n = 56]) | AGA: neonates with a birthweight >10th centile (n = 78) | Cases: 31.7 weeks (early-onset IUGR) and 38.3 weeks (late-onset IUGR) Controls: 31.5 weeks and 38.7 weeks | Cord blood; at birth | Untargeted | ¹H NMR | Early-onset IUGR: unsaturated lipids; lipid VLDL; triglycerides; acetone; creatine; glutamine Late-onset IUGR: unsaturated lipids; lipid VLDL; leucine | Early-onset IUGR: glucose; choline; phenylalanine Late-onset IUGR: choline; glutamine; tyrosine; valine; alanine |
| Dessì[35], 2014 (Italy) | IUGR: newborns with "IUGR diagnosed ultrasonographically in the prenatal period" and birthweight <10th centile (n = 12) | AGA: newborns with a birthweight between the 10th and 90th centiles (n = 17) | Cases: 37.9 weeks Controls: 38.3 weeks | Neonatal urine; within 8 h of birth and before the first feed | Untargeted | ¹H NMR | Citrate; creatinine; creatine; myo-inositol; betaine/trimethylamine-N-oxide; glycine | Urea, aromatic compounds, and branched chain amino acids |
| Maitre[36], 2014 (Greece) | FGR: women who subsequently delivered neonates with a birthweight <10th customised centile (n = 36); SGA, unreported (n = 19) | Unreported (n = 275) | Cases: 39.0 weeks (FGR) and 38.8 weeks (SGA) Controls: 39.0 weeks | Maternal urine: at 11–13 weeks' gestation | Untargeted | ¹H NMR | FGR: none SGA: none | FGR: acetate; formate; tyrosine; trimethylamine SGA: none |
| Sulek[37], 2014 (Singapore) | SGA: women who subsequently delivered neonates with a birthweight <10th centile (n = 41) | Women who subsequently delivered "appropriately grown" neonates (n = 42) | Cases: 39.0 weeks Controls: 39.0 weeks | Maternal hair: at 26–28 weeks' gestation | Untargeted | GC-MS | NADP_NADPH; palmitate; 2-methy-loctadecanoate; myristate; margarate; stearate; dodecanoate; octanoate; heptadecane; nicotinamide | 3-hydroxybenzoate; levulinate; 1-aminocyclopropane1-carboxylate; citraconate; lactate; glycine; proline; isoleucine; serine; leucine; glutamate; phenylalanine; alanine; valine; aspartate; threonine; tyrosine; methionine; lysine; pyroglutamate; ornithine; glutathione |
| Liu[38], 2016 (China) | IUGR: neonates with a birthweight <10th centile (n = 60: birthweight <3rd centile [n = 25]; birthweight between the 3rd and <5th centile [n = 20]; birthweight between the 5th and <10th centile [n = 15]) | AGA: neonates with a birthweight between the 10th and 90th centile (n = 60) | Cases: 36.8 weeks (birthweight <3rd centile), 35.6 weeks (birthweight 3rd to <5th centile), and 35.5 weeks (birthweight 5th to <10th centile) Controls: 35.9 weeks | Dried blood spots (heel prick) for newborn metabolic screening; 3–7 days after birth | Targeted | HPLC-MS | IUGR with a birthweight <3rd centile: homocysteine IUGR with a birthweight between the 3rd and <5th centile: homocysteine; ornithine; isovaleryl carnitine IUGR with a birthweight between the 5th and <10 centile: none | IUGR with a birthweight <3rd centile: methionine; ornithine; serine; tyrosine IUGR with a birthweight between the 3rd and <5th centile: none IUGR with a birthweight between the 5th and <10 centile: none |
| Abd El-Wahed[39], 2017 (Egypt) | SGA: neonates with a birthweight <10th centile (n = 40) | AGA: neonates with a birthweight between the 10th and 90th centiles (n = 20) | Cases: 34.0 weeks Controls: 35.0 weeks | Cord blood; at birth | Untargeted | UPLC-MS | C18-OH; C16-OH; carnitine; arginine; aspartic; valine; alanine; leucine; isoleucine; glutamic acid; tyrosine; ornithine; phenylalanine; citrulline | Histidine; methionine |

**Table 1 (continued) | Main characteristics and findings of studies included in the systematic review**

| First author, year (Country) | Case definition[a] (n) | Control definition[a] (n) | Mean or median gestational age at birth | Biological sample; sampling time | Metabolomics approach | Analytical platform used | Significantly up-regulated metabolites | Significantly down-regulated metabolites |
|---|---|---|---|---|---|---|---|---|
| Visentin[40], 2017 (Italy) | (1) IUGR: EFW <3rd centile without Doppler abnormalities and birthweight <3rd centile or EFW <10th centile with Doppler abnormalities and birthweight <10th centile (n = 11); (2) SGA: EFW <10th centile without Doppler abnormalities and birthweight <10th centile (n = 10) | AGA: EFW between the 10th–90th centiles and birthweight between the 10th and 90th centiles at term (n = 12) | Cases: 36.0 weeks (IUGR) and 37.5 weeks (SGA) Controls: 38.5 weeks | Maternal plasma; "soon after birth, at hospitalization" Cord blood; at birth | Targeted | GC-MS | Maternal blood: (1) IUGR: none; (2) SGA: C8:0; C10:0; C12:0; C18:0 Cord blood: (1) IUGR: C6:0; C8:0; C10:0; C12:0; (2) SGA: C6:0; C8:0; C10:0; C12:0; C16:0; C18:0 | Maternal blood: (1) IUGR: none; (2) SGA: none Cord blood. (1) IUGR: none; (2) SGA: none |
| Delplancke[41], 2018 (New Zealand) | SGA: neonates with a birthweight <10th customised centile (n = 20) | Neonates from "healthy pregnancies" (n = 73) | Cases: 39.1 weeks Controls: 39.9 weeks | Maternal hair; at second and third trimester | Untargeted | GC-MS and LC-MS | At second trimester: margaric acid; pentadecanoic acid; myristic acid At third trimester: none | At second trimester: none At third trimester: none |
| Lu[42], 2018 (Germany) | SGA: neonates with a birthweight <10th centile (n = 23) | AGA: neonates with a birthweight between the 10th and 97th centiles (n = 198) | 39.0 weeks (entire cohort) | Maternal serum; during labour prior to birth Cord blood; at birth | Targeted | FIA-ESI-MS/MS | Maternal blood: none Cord blood: none | Maternal blood: none Cord blood: lysoPC (14:0); lysoPC (16:1); lysoPC (18:1) |
| Miranda[43], 2018 (Spain) | Term neonates with a birthweight <10th centile (n = 52: FGR [birthweight <3rd centile and/or abnormal uterine artery Doppler and/or abnormal cerebroplacental ratio; n = 27] and SGA [birthweight between the 3rd-9th centiles and normal fetoplacental Doppler; n = 25]) | AGA: term neonates with a birthweight between the 20th and 90th centiles (n = 28) | Cases: 37.8 weeks (FGR) and 39.4 weeks (SGA) Controls: 39.8 weeks | Maternal plasma; 2–4 h after birth Cord blood; at birth | Untargeted and targeted | ¹H NMR | Maternal blood: (1) IUGR: none; (2) SGA: none Cord blood: (1) IUGR: cholesterol VLDL and IDL; triglycerides VLDL and IDL; large, medium and small VLDL particle types; medium LDL particle types; large HDL particle types; PCs (area peak 1; height peak 1); glycoproteins (area peak 2; height peak 2; width peak 2; width peak 3); acetate; formate; (2) SGA: formate | Maternal blood: (1) IUGR: triglycerides-HDL; large and medium HDL particle types; PCs (width peak 1): alanine; citrate; 2-oxoisovaleric acid; pyruvate; (2) SGA: cholesterol-IDL; triglycerides IDL and HDL; citrate; 2-oxoisovaleric acid Cord blood: (1) IUGR: none; (2) SGA: none |
| Bahado-Singh[44], 2019 (United States) | Suspected IUGR: neonates with a birthweight <10th centile (n = 39) | AGA: neonates with a birthweight ≥10th centile (n = 39) | Cases: unreported Controls: unreported | Cord blood; at birth | Untargeted and targeted | DI-LC-MS/MS and ¹H NMR | Threonine; DOPA; kynurenine; lysoPC a C16:1; lysoPC a C18:1; lysoPC a C18:2; lysoPC a C20:3; PC aa C38:3 | Creatinine; C0; C10:1; C12:1; C2; C4; PC.aa.C24.0; PC.aa.C26.0; PC aa C32:0; PC aa C36:4; PC aa C38:4; PC aa C40:4; PC ae C36:0; PC ae C36:3; PC ae C36:5; PC ae C38:4; choline |
| Alfano[45], 2020 (Belgium) | SGA: neonates with a birthweight <10th centile (n = 14) | AGA: neonates with a birthweight between the 10th and 90th centiles (n = 155) | 39.1 weeks (entire cohort) | Cord blood; at birth | Untargeted | UHPLC-QTOF-MS | None | None |
| Briana[46], 2020 (Greece) | IUGR: neonates with a birthweight ≤10th customised centile (n = 19) | AGA: term neonates with a birthweight between the 11th and 89th customised centiles (n = 60) | Cases: 38.0 weeks Controls: 39.0 weeks | Milk; third to fourth day postpartum | Untargeted | ¹H NMR | N-acetylglutamine; citric acid; choline; phosphocholine; lactose | Valine; isoleucine |

**Table 1 (continued) | Main characteristics and findings of studies included in the systematic review**

| First author, year (Country) | Case definition[a] (n) | Control definition[a] (n) | Mean or median gestational age at birth | Biological sample; sampling time | Metabolomics approach | Analytical platform used | Significantly up-regulated metabolites | Significantly down-regulated metabolites |
|---|---|---|---|---|---|---|---|---|
| Clinton[47], 2020 (United States) | FGR: term neonates with a birthweight <10th centile (n = 30) | Non-FGR: term neonates with a birthweight ≥10th centile (n = 30) | Cases: 38.0 weeks Controls: 38.5 weeks | Maternal urine; at 10–26 weeks' gestation | Untargeted | GC/EI-MS | At 10 weeks: benzoic acid; malonic acid; 2-ketoleucine/ketoisoleucine; 2-ketobutyric acid; 2-methylglutaric acid; and acetoacetate At 26 weeks: 1,2-propanediol; kynurenic acid; n-heptanoic acid; and benzoic acid | None |
| Kan[48], 2020 (Russia) | "Confirmed diagnosis of IUGR", (n = 17) | "Healthy women with uncomplicated pregnancy" (n = 21) | Cases: 36.1 weeks Controls: 39.2 weeks | Maternal plasma; "at delivery" | Targeted | HPLC-MS | Aspartate; beta-alanine; carnosine; gamma aminobutyrate; methionine; ornithine; tryptophan; alanine; glutamine; glycine; histidine; isoleucine; lysine; phenylalanine; serine | Asparagine; cystine; O-phosphoryl-ethanolamine |
| Sovio[49], 2020 (United Kingdom) | FGR: term neonates with a birthweight <3rd customised centile or birthweight between the 3rd and <10th customised centiles combined with the lowest decile of fetal abdominal growth velocity (n = 175) | AGA: term neonates with a customised birthweight ≥10th centile (n = 299) | Cases: 40.1 weeks Controls: 40.3 weeks | Maternal serum; at 12, 20, and 28 weeks' gestation | Untargeted | UPLC–MS/MS | At 20 or 28 weeks' gestation: 1-(1-enyl-stearoyl)-2-oleoyl-GPC (P18:0/18:1); 1-(1-enyl-palmitoyl)-2-oleoyl-GPC (P16:0/18:1); 1,5-anhydroglucitol; cotinine N-oxide; 4-androsten-3beta,17-beta-diol monosulfate; 1-(1-enyl-palmitoyl)-2-palmitoleoyl-GPC (P-16:0/16:1); hydroxycotinine; acisoga; 3-hydroxycotinine glucuronide; O-cresol sulphate; dehydroisoandrosterone sulphate | At 20 or 28 weeks' gestation: 5alpha-androstan-3alpha,17alpha-diol disulfate; estriol 3-sulphate; 4-cholesten-3-one; pregnanolone/allopregnanolone sulphate; 5alpha-pregnan-3alpha,20beta-diol disulfate 1; N1,N12-diacetylspermine; 17alpha-hydroxypregnanolone glucuronide; 5alpha-pregnan-3beta,20beta-diol monosulfate; progesterone; pregnanediol-3-glucuronide |
| Welch[50], 2020 (United States) | SGA: neonates with a birthweight ≤10th centile (n = 31) | AGA: neonates with a birthweight between the 10th and 90th centiles (n = 31) | Cases: 37.9 weeks Controls: 39.0 weeks | Maternal plasma; at 11, 25, and 35 weeks' gestation | Targeted | LC-MS/MS | 5,6-DHET; 8,9-DHET; 14,15-DHET; 8-HETE; 12-HETE; 15-HETE | None |
| Beken[51], 2021 (Türkiye) | SGA: exclusively breastfed term neonates with a birthweight <10th centile (n = 69) | AGA: exclusively breastfed term neonates with a birthweight between the 25th and 75th centiles (n = 168) | Cases: 39.0 weeks Controls: 39.0 weeks | Dried blood spots (heel prick) for newborn metabolic screening; 24–48 h after birth | Targeted | LC-MS/MS | Alanine; methionine; phenylalanine; leucine/isoleucine; glycine; free carnitine; acetyl carnitine; butyryl carnitine; isovaleryl carnitine; decenoyl carnitine; oleyl carnitine; linolenoyl carnitine; 3-OH tetradecanoyl carnitine; 3-OH linonenoyl carnitine | Propionyl carnitine; methylglutaryl carnitine |
| Byeon[52], 2021 (Bangladesh) | SGA: neonates with a birthweight <3rd centile (n = 20) | AGA: neonates with a birthweight ≥50th centile (n = 20) | Cases: 40.1 weeks Controls: 39.7 weeks | Maternal serum; at 24–28 weeks' gestation | Untargeted | UHPLC-MS/MS | None | LPA (20:4) |
| Morillon[53], 2021 (Ireland and New Zealand) | SGA: neonates with a birthweight <10th customised centile (n = 80) | "Healthy and uncomplicated pregnancies" (median customised centile, ~52nd) (n = 80) | Cases: 39.7 weeks Controls: 40.4 weeks | Maternal plasma (lipidomics); at 20 weeks' gestation Maternal urine (metabolomics); at 20 weeks' gestation | Untargeted | UPLC-QTOF-MS | Maternal blood: PE (P-31:0); PE (42:1); PE (36:4); PS (O-37:0); PS (41:5); PS (37:2); PS (43:6); PS (P-34:0); PC (O-42:4); PC (40:5); PC (38:6); LysoPC (16:0); PA (O-36:2); LysoPA (18:1); PI (37:1); PI (P-33:1); PGP (38:4); PGP (40:4); PG (36:6); PG (39:8); PG (38:4); DG (44:4); DG (O-34:1); N,N-dimethyl arachidonoyl amine; N-palmitoyl valine; SM (34:1); ganglioside GA2 (40:1); Cer (34:0); Cer (39:2) Maternal urine: None | Maternal blood: CL(72:2); TG(64:15); 8S-hydroxy-hexadecanoic acid; CE(17:0) Maternal urine: sulfolithocholic acid; estriol-16-glucuronide; D-glucuronic acid; neuromedin N (1–4); 4-hydroxybenzaldehyde; 18-hydroxycortisol; beta-1,4-mannosyl-N-acetylglucosamine |

**Table 1 (continued) | Main characteristics and findings of studies included in the systematic review**

| First author, year (Country) | Case definition[a] (n) | Control definition[a] (n) | Mean or median gestational age at birth | Biological sample; sampling time | Metabolomics approach | Analytical platform used | Significantly up-regulated metabolites | Significantly down-regulated metabolites |
|---|---|---|---|---|---|---|---|---|
| Moros[54], 2021 (Greece) | IUGR: term neonates with a birthweight ≤10th customised centile (n = 41) | AGA: term neonates with a birthweight between the 10th and 90th customised centiles (n = 36) | Cases: 38.8 weeks Controls: 39.7 weeks | Maternal serum; 2–4 h after birth Cord blood; at birth | Untargeted | ¹H NMR | Maternal blood: alanine; leucine; iso-leucine; valine; 3-hydroxybutyrate Cord blood: alanine; leucine; iso-leucine; valine | Maternal blood: phenylalanine; glycerol Cord blood: phenylalanine; gly-cerol; tryptophan |
| Schupper[55], 2021 (Israel) | SGA: term neonates with a birthweight <10th centile (n = 6380: severe SGA [<3rd centile; n = 1391] and moderate SGA [between the 3rd and <10 centiles; n = 4989]) | AGA: term neonates with a birthweight between the 10th and 90th centiles (n = 61,068) | 39.0 weeks (entire cohort) | Dried blood spots (heel prick) for neonatal metabolic screening; 36–72 h after birth | Targeted | UPLC-MS | All SGA: alanine; methionine; proline; total carnitine; free carnitine Severe SGA: alanine; leucine; proline; ornithine; methionine; free carnitine | All SGA: valine Severe SGA: valine; glutamine |
| Youssef[56], 2021 (Spain) | FGR: EFW and birth-weight <10th centile associated with either abnormal CPR (<5th centile) or abnormal uterine artery pulsatility index >95th centile), or birthweight <3rd cen-tile (n = 44) | AGA: full-term neo-nates with EFW and birthweight >10th centile (n = 88) | Cases: 37.6 weeks Controls: 39.6 weeks | Maternal plasma; within 2 h of birth Cord blood; at birth | Untargeted and targeted | ¹H NMR | Maternal blood: None Cord blood: triglycerides IDL; choles-terol IDL; choline compound Peak 2 | Maternal blood: None Cord blood: cholesterol HDL; isoleucine |
| Bahado-Singh[57], 2022 (United States) | Suspected FGR: neo-nates with a birthweight <10th centile (n = 19) | AGA: neonates from "uncomplicated term pregnancies" with a birthweight ≥10th centile (n = 30) | Cases: 36.4 weeks Controls: 39.8 weeks | Placenta; within 20 min of birth | Untargeted and targeted | DI-LC-MS/ MS and ¹H NMR | 3-hydroxyisovaleric acid; citric acid; putrescine | Citrulline; ornithine; asymmetric dimethylarginine; alpha-amino adi-pic acid; cis-4-hydroxyproline; creatinine; dihydrox-yphenylalanine; kynurenine; methionine sulfoxide; sarcosine; spermidine; spermine; trans-4-Hydroxyproline; symmetric dime-thylarginine; carnitine; C102; C12; C12-DC; C14; C141; C141-OH; C142-OH; C16; C16-OH; C161; C161-OH; C162; C162-OH; C18; C181; C181-OH; C182; C3;C3-OH; C3-DC; C4-OH; C41; C5-M-DC; C5-OH; C3-DC-M; C51; C9; lysoPC a C140; lysoPC a C160; lysoPC a C161; lysoPC a C170; lysoPC a C180; lysoPC a C181; lysoPC a C182; lysoPC a C203; lysoPC a C204; lysoPC a C240; lysoPC a C260; lysoPC a C261; lysoPC a C280; lysoPC a C281; PC aa C240; PC aa C260; PC aa C281; PC aa C300; PC aa C320; PC aa C321; PC aa C322; PC aa C323; PC aa C341; PC aa C342; PC aa C343; PC aa C344; PC aa C360; PC aa C361; PC aa C362; PC aa C363; PC aa C364; PC aa C365; PC aa C366; PC aa C380; PC aa C383; PC aa C384; PC aa C385; PC aa C386; PC aa C401; PC aa C402; PC aa C403; PC aa C404; PC aa C405; PC aa |

**Table 1 (continued) | Main characteristics and findings of studies included in the systematic review**

| First author, year (Country) | Case definition[a] (n) | Control definition[a] (n) | Mean or median gestational age at birth | Biological sample; sampling time | Metabolomics approach | Analytical platform used | Significantly up-regulated metabolites | Significantly down-regulated metabolites |
|---|---|---|---|---|---|---|---|---|
| | | | | | | | | C406; PC aa C420; PC aa C421; PC aa C422; PC aa C424; PC aa C425; PC aa C426; PC ae C300; PC ae C301; PC ae C302; PC ae C321; PC ae C322; PC ae C340; PC ae C341; PC ae C342; PC ae C343; PC ae C360; PC ae C361; PC ae C362; PC ae C363; PC ae C364; PC ae C365; PC ae C380; PC ae C381; PC ae C382; PC ae C383; PC ae C384; PC ae C385; PC ae C386; PC ae C401; PC ae C402; PC ae C403; PC ae C404; PC ae C405; PC ae C406; PC ae C420; PC ae C421; PC ae C422; PC ae C423; PC ae C424; PC ae C425; PC ae C443; PC ae C444; PC ae C445; PC ae C446; SM OH C141; SM OH C161; SM OH C221; SM OH C222; SM OH C241; SM C160: SM C161; SM C180; SM C181; SM C202; SM C240; SM C241; SM C260; SM C261; H1: 1-methylhistidine; acetic acid; ascorbic acid; choline; D-glucose; glycine; hypoxanthine; tyrosine; L-alanine; L-proline; L-threonine; L-asparagine; isoleucine; L-lysine; L-serine; aspartate; myo-inositol; taurine; succinate; pyroglutamic acid; urea; uracil; 3-hydroxybutyric acid; 2-hydroxyisovalerate; L-arginine; L-glutamine; L-leucine; hippuric acid; isopropyl alcohol; valine; acetone |
| Chao de la Barca[58], 2022 (France) | IUGR: neonates with a birthweight <10th centile, reduction of fetal growth on ultrasound, and the presence of a notch in at least one uterine artery and abnormalities on umbilical artery and/or cerebral artery and/or ductus venosus on Doppler ultrasound during pregnancy (n = 15) | Neonates with a birthweight ≥10th centile from normal pregnancies who underwent a planned caesarean section before labour at term (n = 15) | Cases: 35.2 weeks Controls: 39.1 weeks | Cord blood; at birth | Targeted | LC-MS/MS and FIA-MS/MS | Alanine; asparagine; tyrosine; glutamine; proline; C0; C2, C4; PC aa C24:0, PC aa C32:0; trans-4-hydroxyproline; alpha-aminoadipic acid; spermine; LysoPC a C26:1 | LysoPC a C16:0; lysoPC a C16:1; lysoPC a C17:0; lysoPC a C18:0; lysoPC a C18:1; lysoPC a C18:2; lysoPC a C20:3; lysoPC a C20:4); PC aa C36:0; PC aa C36:1; PC aa C36:3; PC aa C36:6; PC aa C38:0; PC aa C38:3; PC aa C38:6; PC aa C40:6; PC aa C42:0; PC aa C42:6; PC ae C36:3; PC ae C38:0; PC ae C38:3; PC ae C38:6; PC ae C40:1; PC ae C40:2; PC ae C40:3; PC ae C40:4; PC ae C40:5; PC ae C40:6; PC ae C42:2; PC ae C42:3; PC ae C42:4; PC ae C42:5; PC ae C44:4; PC ae C44:5; PC ae C44:6; SM C24:0; SM C26:0; SM (OH) C22:1; SM (OH) C24:1; tryptophan |

**Table 1 (continued) | Main characteristics and findings of studies included in the systematic review**

| First author, year (Country) | Case definition[a] (n) | Control definition[a] (n) | Mean or median gestational age at birth | Biological sample; sampling time | Metabolomics approach | Analytical platform used | Significantly up-regulated metabolites | Significantly down-regulated metabolites |
|---|---|---|---|---|---|---|---|---|
| Gonzalez-Riano[59], 2022 (Spain) | SGA: term neonates with a birthweight Z-score below −2 SD (n = 12) | AGA: term neonates with a birthweight Z-score between −1 and +1 SD (n = 12) | Cases: 38.6 weeks Controls: 39.6 weeks | Cord blood; at birth | Untargeted | UHPLC-ESI-QTOF-MS | 11-HEDE; 19-hydroxy-PGE2; docosahexaenoic acid; docosapentaenoic acid; hexacosanedioic acid; decanoylcarnitine; decenoylcarnitine; FAHFA(30:1); 9-HODE; 9-OxoODE; methyl-FA 18:3;2OOH; DG 16:0/18:2/0:0; DG 18:1/18:1/0:0; DG 18:1/18:2/0:0; DG 18:2/16:0/0:0; DG 18:3/19:0/0:0; TG 14:0/18:1/22:6; TG 14:0/22:2/18:2; TG 14:0/22:4/18:2; TG 15:0/18:3/19:0; TG 15:0/20:0/0-18:0; TG 16:0/16:0/20:4; TG 16:0/16:1/20:4; TG 16:0/16:1/22:6; TG 16:0/17:2/22:2; TG 16:0/18:0/18:3; TG 16:0/18:1/20:4; TG 16:0/18:1/22:6; TG 16:0/18:2/20:4; TG 16:0/18:2/22:6; TG 16:0/18:3/22:5; TG 16:0/20:1/O-18:0; TG 16:0/20:4/22:6; TG 16:1/18:1/18:1; TG 16:1/18:1/18:2; TG 16:1/18:1/20:2; TG 16:1/18:2/20:3; TG 16:0/20:5; 22:6; TG 18:0/18:3/18:4; TG 18:0/18:3; 20:4; TG 18:1/18:1/18:2; TG 18:1/18:2; 18:1; TG 18:1/18:2/18:4; TG 18:1/18:3; 18:3; TG 18:1/18:2/22:0; TG 20:4/16:0/O-18:0; PC O-22:1/22:3; PC 16:0/18:2;OH; PC 16:0/20:4;5OH; PC 18:0/18:2;OH; PC 18:0/20:4;12OH; PS 22:0/20:0; PG O-18:0/18:0; a-L-fucopyranosyl-(1->2)-b-D-galactopyranosyl-(1->2)-D-xylose; SM 18:0;O2/26:0; 32-oxolanosterol; 22:0-Glc-cholesterol; 22:2-Glc-cholesterol; DiMe(9,3) cholesterol ester; cholesteryl 11-hydroperoxy-eicosatetraenoate; cholesteryl-6-O-myristoyl-alpha-D-glucoside; sitostanyl:18:0; 9-HODE cholesteryl ester; 3-Deoxyvitamin D3 | PC O-18:1/18:2; lysoPC 14:0/0:0; lysoPC 16:1/0:0; lysoPC 18:1/0:0; lysoPC 20:3/0:0; PC 14:1/20:2; PC 16:0/18:1; PC 16:0/20:3; PC 16:0/22:4; PC 16:0/16:1; PC 16:0/18:1; PC 16:0/20:3; PC 18:0/18:1; PC 18:0/20:5; PC 18:1/18:2; PC 18:2/22:2; PC 22:2/16:1; PC18:0/20:2; PC 22:4/18:1; TG 80:5; TG 84:5; PS 20:0/22:6; PS 22:0/16:0; 2-O-[alpha-D-glucopyranosyl-(1->6)-alpha-D-glucopyranosyl]-D-glycerate; caffeine; SM 18:1;O2/21:0; SM 18:1;O2/20:0; SM 18:0;O2/22:0; CE 16:1; CE 18:3; CE 20:3 |
| Karaer[60], 2022 (Türkiye) | FGR: neonates born after 32 weeks' gestation with an EFW or AC <3rd centile (n = 10) | Neonates born after 32 weeks' gestation who did not meet the above criteria (n = 14) | Cases: 36.5 weeks Controls: 38.9 weeks | Placenta; at birth | Untargeted | HR-MAS and NMR | Lactate; glutamine; glycerophosphocholine; phosphocholine; taurine; myoinositol | Unreported |
| Liu[61], 2022 (China) | SGA: term neonates with a birthweight <10th centile (n = 713) | AGA: term neonates with a birthweight between the 10th and 90th centiles (n = 7866) | Cases: 39.2 weeks Controls: 39.0 weeks | Dried blood spots (heel prick) for newborn metabolic screening; on third day of life | Targeted | HPLC-MS | Alanine; citrulline; ornithine; proline; free carnitine; total carnitine; acetylcholine; butyryl carnitine; octanoyl carnitine; decanoyl carnitine; decenoyl carnitine; dodecanoyl carnitine; dodecenoyl carnitine; myristoyl carnitine; myristoleyl carnitine; tetradecadienoyl carnitine; palmitoyl carnitine; hexadecenoyl carnitine; 3-hydroxy(OH) palmitoleyl carnitine; octadecanoyl carnitine; octadecenoyl carnitine; linoleyl carnitine; 3-hydroxy(OH)octadecenoyl carnitine | Propionyl carnitine |

**Table 1 (continued) | Main characteristics and findings of studies included in the systematic review**

| First author, year (Country) | Case definition[a] (n) | Control definition[a] (n) | Mean or median gestational age at birth | Biological sample; sampling time | Metabolomics approach | Analytical platform used | Significantly up-regulated metabolites | Significantly down-regulated metabolites |
|---|---|---|---|---|---|---|---|---|
| McCarthy[62], 2022 (United States) | SGA: neonates with a birthweight <10th centile (n = 147,287) | AGA: neonates with a birthweight between the 10th and 90th centiles (n = 589,148) | Cases: unreported Controls: unreported | Dried blood spots (heel prick) for neonatal metabolic screening; between 12 h and 8 days after birth | Targeted | LC-MS | 17-hydroxyprogesterone; thyroid stimulating hormone; galactose-1-phosphate uridyl transferase; 5-oxo-proline; alanine; citrulline; glycine; methionine; ornithine; phenylalanine; proline; free carnitine; C2; C3-DC; C4; C5; C5-DC; C5-OH; C6; C8; C8:1; C10; C10:1; C12; C14; C18; C18:1; C18:2 | Arginine; valine; C3; C5:1; C14-OH; C16:1; C16-OH; C18-OH |
| Umeda[63], 2022 (Japan) | SGA: neonates with a birthweight <10th centile (n = 11) | AGA: neonates with a birthweight ≥10th centile (n = 179) | 39.2 weeks (entire cohort) | Cord blood; at birth | Targeted | LC-MS/MS | diHOME; 9,10-diHOME; 12,13-diHOME; 14,15-diHETE | 5,6-diHETrE |
| Voerman[64], 2022 (The Netherlands) | SGA: neonates with a birthweight <10th centile (n = 98) | AGA: neonates with a birthweight between the 10th and 90th centiles (n = 780) | 40.3 weeks (entire cohort) | Maternal serum; at a median gestational age of 12.8 weeks (95% range, 9.9–16.9 weeks) Cord blood; at birth | Targeted | HPLC-MS | Maternal blood: None Cord blood: NEFA C26:2 | Maternal blood: None Cord blood: PC.aa C36:3; lysoPC.a C14:0; lysoPC.a C16:0; lysoPC.a C16:1; lysoPC.a C18:1; lysoPC.a C18:2; lysoPC.a C18:3; lysoPC.a C20:3; lysoPC.a C20:4; lysoPC.a C22:6; lysoPC.e C18:1; SM.a C34:2; SM.a C38:3 |
| Bartho[65], 2023 (Australia) | FGR: neonates with a birthweight <5th centile (n = 55) | Unreported; median birthweight, 3589 g (n = 72) | Cases: unreported Controls: unreported | Maternal plasma; at 36 (35^+0–37^+0) weeks' gestation | Targeted | LC-QQQ-MS | CE 15:0; CE 16:1; CE 17:1; CE 22:4; CE 24:6; Cer(d20:1/24:1) | None |
| Chen[66], 2023 (China) | FGR: EFW or AC <10th centile (n = 18) | "Normal fetal growth" with normal fetal chromosome karyotype (n = 10) | Cases: 36.8 weeks Controls: 38.6 weeks | Amniotic fluid; at 30.1 ± 3.4 weeks (cases) and 19.1 ± 1.6 weeks' gestation (controls) | Untargeted | GS-MS | Amniotic fluid supernatant: hexadecane acid; 2-hydroxypyridine; octadecanoic acid; urea; 2-hydroxyisobutyric acid; ethanolamine; glycerol; D-glycerate; xylitol; butane 1,2,3,4-tetraol; maleic acid; 2-oxyglutaric acid; D(+)-cellulose; hydroxyacetone Amniotic fluid cell sediment: glycolic acid; malic acid; 2-keto-L-gluconate; malt dust; D-glycerate; maleic acid; butane 1,2,3,4-tetraol; threitol; D-(+)-cellulose | Amniotic fluid supernatant: l-glutamic acid; leucine; phenylalanine; isoleucine; valine; diisopropylamine; isothreonine; proline; DL-alanine; L-alanine; 4-hydroxyproline; N-methyl-D-L-alanine; glycine Amniotic fluid cell sediment: l-glutamic acid; phosphoric acid; L-methionine S-oxide; L-valine; L-alanine; l-leucine; DL-alanine; DL glyceraldehyde; N-methyl-D-L-alanine; 2(methylamino) ethanol; butylamine |
| Elhakeem[67], 2023 (United Kingdom and Australia) | SGA: neonates with a birthweight <5th centile (n = 227) | AGA: neonates with a birthweight between the 5th and 95th centiles (n = 2998) | Cases: unreported Controls: unreported | Cord blood; at birth | Targeted | ¹H NMR | Total VLD cholesterol; total very small VLDL lipids; very small VLDL particles; omega-3 fatty acids | Total HDL2 cholesterol; total HDL3 cholesterol; total medium-sized HDL lipids; medium-sized HDL particles; apolipoprotein A-I; histidine |
| Jafri[88], 2023 (Pakistan) | SGA: neonates with a birthweight <10th centile (n = 219) | AGA: neonates with a birthweight between the 10th and 90th centiles (n = 391) | 38.7 weeks (entire cohort) | Dried blood spots (heel prick) for neonatal metabolic screening; 48–72 h after birth | Targeted | LC-QQQ-MS | None | Alanine; arginine; citrulline; ornithine; C3-DC; C4-OH; C5-DC; C6; C8; C8:1; C10; C10:1; C12:1; C14:2 |
| Priante[69], 2023 (Italy) | IUGR: neonates born <32 weeks' gestation with an EFW or AC <3rd centile or <10th centile plus uterine or umbilical artery pulsatility index >95th centile (n = 15) | Non-IUGR: neonates born <32 weeks' gestation who did not meet the above criteria (n = 19) | Cases: 30.1 weeks Controls: 29.6 weeks | Neonatal urine; within 48 h of life | Untargeted | UPLC-MS | 3-Indolepropionic acid; L-tryptophan; L-histidine; L-cysteine; androstenedione; 7alfa-hydroxydehydroepiandrosterone; N-butyrylglycine; L-2-aminobutyric acid; isovalerylglucuronide; N-acetylcystathionine; 3-(3,4- | 3-hydroxyanthranilic acid; aspartylglycosamine; carnosine; inosine; dihydrocortisone; L-methionine; dihydrocortisol; 5beta-dihydrocortisol; 7-ketodeoxycholic acid; 5-hydroxyindoleacetic acid; pantetheine; N-a-acetylcitrulline; |

**Table 1 (continued) | Main characteristics and findings of studies included in the systematic review**

| First author, year (Country) | Case definition[a] (n) | Control definition[a] (n) | Mean or median gestational age at birth | Biological sample; sampling time | Metabolomics approach | Analytical platform used | Significantly up-regulated metabolites | Significantly down-regulated metabolites |
|---|---|---|---|---|---|---|---|---|
| | | | | | | | dihydroxyphenyl)lactic acid; 3,4-dimethylbenzoic acid; 3-sialyl-N-acetyllactosamine; aspartylysine; gamma glutamyl ornithine; 2,2-dimethylsuccinic acid; estrone; nicotinamide ribotide; 3-hydroxysebacic acid; 3-hexenedioic acid; 3-methylglutaconic acid; cyclic GMP; GMP; purine | indolelactic acid; 5-hydroxykynurenamine; ascorbic acid; fumaric acid; 5-Amino-2-oxopentanoate; 4-methylcatechol; tyrosol; pantetheine; glutamylphenylalanine; N-a-acetylcitrulline; normetanephrine; methylnoradrenaline; 2-methylbutyrylglycine; N-acetylneuraminic acid; O-desmethylangolensin; 1-methylguanosine; D-glucuronic acid; methylhippuric acid; 7,9-dimethyluric acid; N-acetylglutamic acid; dethiobiotin; N-acetylaspartylglutamic acid; 2-methylbenzoic acid; deoxycorticosterone; 16b-hydroxyestrone; hydroxykynurenine; neuraminic acid; umanopterin; 8-hydroxy-deoxyguanosine; glucosylceramide (d18:1/26:1(17Z); TG (16:1(9Z)/16:1(9Z)/18:2(9Z,12Z) [iso3]; 5,6-dihydrouridine; aspartyl-L-proline; dethiobiotin |
| Tao[70], 2023 (China) | FGR: EFW or AC <10th centile and placental disorders or umbilical cord abnormalities by postnatal confirmation (n = 54:19 maternal blood and 35 maternal faeces) | Normal term deliveries, with EFW between the 10th–90th centiles and birthweight between 2500 and 4000 g (n = 66: 31 maternal blood and 35 maternal faeces) | Cases: 37.2 weeks (maternal blood) and 38.0 weeks (maternal faeces) Controls: 39.8 weeks (maternal blood) and 39.5 weeks (maternal faeces) | Maternal serum (n = 50); in the "third trimester" Maternal faeces (n = 70); in the "third trimester" | Untargeted | UHPLC-MS/MS | Maternal blood: imidazole-4-acetic-acid; pentadecan-1-ol; pinitol; allantoin; lyxonic acid; 3-oxalo-malic acid; glycolic acid-2-phosphate; octadecadienoic acid; nicotinic acid Maternal faeces: p-synephrine; N,N-diethylbenzeneacetamide; marmesin rhamnoside; N1-methyl-4-piridone-3-carboxamide; lysoPC (18:2(9Z,12Z)); dehydrophytosphingosine; 4,6-dihydroxyquinoline | Maternal blood: dodecanoic acid; malic acid; 9-hexadecenoic acid; maltose Maternal faeces: (3alphaOH,20S,24S)-3,19:20,24-diepoxydammarane-3,25-diol; eremopetasitenin C3; glyceollin IV; 6-(2-carboxyethyl)-7-hydroxy-2,2-dimethyl-4-chromanone glucoside; physagulin E(2S,4 R,5S)-muscarine; ginkgolide C; D-erythro-eritadenine; histidinyl-valine; 6-epi-7-isocucurbic acid glucoside; pyrraline; 2,5-dimethyl-1H-pyrrole |
| Troisi[71], 2023 (United States) | SGA: neonates with a birthweight <10th centile (n = 118) | AGA: neonates with a birthweight between the 10th and 89th centiles (n = 326) | Cases: 37.1 weeks Controls: 37.7 weeks | Placenta; at birth | Untargeted | GC-MS | Asparagine; glycerophosphocholine; aspartic acid; tyrosine; isoleucine; erythritol; serine; deoxyribose; lactic acid | Taurine; glycine |
| Yang[72], 2023 (China) | FGR: EFW or AC <10th centile with oligohydramnios and abnormal umbilical artery flow, delivered by caesarean section without labour (n = 35) | Full-term normal pregnancies, delivered by caesarean section without labour (n = 35) | Cases: 38.0 weeks Controls: 39.5 weeks | Maternal serum; 24 h before caesarean section Placenta; at birth Cord blood; at birth | Untargeted | GC-MS | Maternal blood: 2-oxovaleric acid; DL-gamma-methyl-ketoglutaramate; lysine; serine; N-(carboxymethyl)-L-alanine; creatinine; glutaric acid; 2-aminoadipic acid; 4-aminobutyric acid; 2-hydroxycinnamic acid; hexanoic acid (C6_0); azelaic acid; 2-oxoglutaric acid Placenta: pentanoic acid, 4-oxo-, methyl ester; L-leucine, methyl ester; DL-phenylalanine, methyl ester; | Maternal blood: dimethyl aminomalonic acid; 3-hydroxydecanoic acid; lactic acid; 2,4-di-tert-butylphenol; phenethyl acetate; palmitic acid (C16_0); 10,13-dimethyltetradecanoic acid (C14_0); margaric acid (C17_0); pentadecanoic acid (C15_0); stearic acid (C18_0); fumaric acid; succinic acid; malic acid; 10,12-octadecadienoic acid (C18_2n-10,12c; trans-vaccenic |

**Table 1 (continued) | Main characteristics and findings of studies included in the systematic review**

| First author, year (Country) | Case definitionª (n) | Control definitionª (n) | Mean or median gestational age at birth | Biological sample; sampling time | Metabolomics approach | Analytical platform used | Significantly up-regulated metabolites | Significantly down-regulated metabolites |
|---|---|---|---|---|---|---|---|---|
| | | | | | | | dimethyl fumarate; octadec-9-en-1-al dimethyl acetal; cholest-5-en-3-ol (3á)-, nonanoate; cholest-5-ene, 3-methoxy-, (3á); 9,12-octadecadienal, dimethyl acetal; methyl pentyl phthalate; iso-butyl methyl phthalate; pentadecane; sulphurous acid, 2-ethylhexil hexyl ester; 1-tetradecene; dibenzo[e,g]ben-zimidazole, 3-ethyl-2-(2-furyl)-; hex-adecanoic acid, 2-hydroxy-methyl ester; eicosanoic acid; heptadecanoic acid, methyl ester; decanoic acid; nonanoic acid methyl ester; non-adecanoic acid; tridecanoic acid: mar-garic acid; succinic acid, 2,2,3,3-tetrafluoropropyl 2-decyl ester; 11-trans-eicosenoic acid, (11E)- C20:1(n-9t); cis-5,8,11-eicosatricenoic acid, methyl ester; 11,14-cis-eicosadienoic; 5,8,11,14,17-eicosapentaenoic acid, methyl ester, all-Z); methyl 5,11,14-eicosatrienoate; methyl linoleate; cis-11,14-eicosadienoic acid, methyl ester; cis-13-octadecenoic acid, methyl ester; 9-cis-hexadecenoic acid; gon-doic acid; gamma-linolenic acid; alpha-linolenic acid Cord blood: trans-4-hydroxyproline; phenylalanine; 2-aminoadipic acid; 2-aminophenylacetic acid | acid; oleic acid (C18_1n-9c); linoleic acid (C18_2n-6,9c; conjugated lino-leic acid (C18_2n-9,11c) Placenta: benzene, 1,2,4-trimethyl-; 2,4-di-tert-butylphenol; 4-(1-methyl-1-silacyclobutyl-1)phenol Cord blood: margaric acid (C17_0); myristic acid (C14_0); pentadeca-noic acid (C15_0); stearic acid (C18_0); gondoic acid (C20_1n-9c); gamma-linolenic acid (C18_3n-6,9,12c); alpha-linolenic acid (C18_3n-3,6,9c); EPA (C:20_5n-3,6,9,12,15c); bishomo-gamma-linolenic acid (C20_3n-6,9,12c); 11,14,17-eicosatrienoic acid (C20_3n-3,6,9c); arachidonic acid (C20_4n-6,9,12,15c); cis-vaccenic acid (C18_1n-7c); 10,12-octadeca-dienoic acid (C18_2n-10,12c); trans-vaccenic acid; oleic acid (C18_1n-9c); linoleic acid (C18_2n-6,9c); conjugated linoleic acid (C18_2n-9,11c) |
| Yeum[73], 2023 (United States) | SGA: neonates with a birthweight ≤10th centile (n = 52: 13 women and 39 newborns) | AGA: neonates with a birthweight between the 11th and 89th centiles (n = 895: 281 women and 614 newborns) | 39.5 weeks (entire cohort of women) and 39.1 weeks (entire cohort of newborns) | Maternal plasma; at 24–28 weeks' gestation (n = 294) Cord blood: at birth (n = 653) | Targeted | LC-MS/MS; FIA-MS/MS | Maternal blood: None Cord blood: hexanoylcarnitine; decanoylcarnitine; dodecanoylcarni-tine; dodecenoylcarnitine; tetra-decanoylcarnitine; tetra-decenoylcarnitine; tetradecadienylcarnitine | Maternal blood: None Cord blood: lysoPC a C16:0; lysoPC a C16:1; lysoPC a C18:0; lysoPC a C18:1; lysoPC a C18:2; lysoPC a C20:3; lysoPC a C20:4; Total lysoPC; Monounsaturated fatty acid/Saturated fatty acid; Total lysoPC /Total PC |
| Zhai[74], 2023 (China) | SGA: term neonates with a birthweight <10th cen-tile (n = 16) | AGA: term neonates with a birthweight at "approximately the 50th centile" (n = 28) | Cases: 38.7 weeks Controls: 38.4 weeks | Maternal plasma; at 37–42 weeks' gestation Cord blood; at birth | Untargeted | UPLC-MS | Maternal blood: PG (16:1/22:6) Cord blood: L-Carnitine | Maternal blood: Cuminaldehyde Cord blood: None |

AC abdominal circumference, AGA appropriate for gestational age, AGA appropriate for gestational age, CE cholesteryl ester, CER ceramide, CL cardiolipin, CPR cerebroplacental ratio, DG diglyceride, DHET dihydroxy-eicosatrienoic acid, DI direct injection, DiHETrE dihydroxyeicosatetraenoic acid, DiHETrE hydroxyeicosatetraenoic acid, DiHOME dihydroxy-octadecenoic acid, DiHOME dihydroxy-octadecenoic acid, EFW estimated fetal weight, EI electron ionization, ESI electrospray ionization, FAHFA fatty acid esters of hydroxy fatty acids, FGR fetal growth restriction, FIA flow injection analysis, GC gas chromatography, GMP guanosine monophosphate, GPC glycerophosphorylcholine, 'H NMR proton nuclear magnetic resonance, HDL high density lipoprotein, HEDE hydroxy-eicosadienoic acid, HETE hydroxyeicosatetraenoic acid, HODE hydro-xyoctadecadienoic acid, HPLC high-performance liquid chromatography, HR-MAS NMR high-resolution magic angle spinning nuclear magnetic resonance, HRMS liquid chromatography high-resolution mass spectrometry, IDL intermediate-density lipoproteins, IUGR intrauterine growth restriction, LC liquid chromatography, LDL low density lipoprotein, LPA lysophosphatidic acid, LysoPC lysophosphatidylcholine, MS mass spectrometry, NADP nicotinamide adenine dinucleotide phosphate, NEFA non-esterified fatty acid, PA phosphatidylglycerophosphate, PC phosphatidylcholine, PE phosphatidylethanolamine, PG phosphatidylglycerol, PGD prostaglandin D, PGE prostaglandin E, PGP phosphatidylglycerophosphate, PI phosphatidylinositol, PS phosphatidylserine, QQQ triple quadrupole, QTOF quadrupole time-of-flight, SGA small for gestational age, SM sphingomyelin, TG triglyceride, UHPLC ultra-high performance liquid chromatography, UPLC ultra performance liquid chromatography, VLDL very low density lipoprotein.

ªAs defined by the authors.

ᵇA total of 574 metabolites showed significant differences in mean levels between SGA and controls at one or more of the oxygen tensions (1%, 6%, and 20%) at which placental villous explants were cultured. 95% of these 574 metabolites showed a lower mean metabolite level in the SGA samples when compared to the controls.

birthweight for gestational age <10th centile plus ultrasound parameters (seven studies[28,30,34,35,43,49,58]); birthweight for gestational age <5th customised (one study[27]) or non-customised (two studies[65,67]) centile; birthweight or estimated fetal weight (EFW) or abdominal circumference (AC) for gestational age <3rd centile (two studies[52,60]); EFW or AC for gestational age <10th centile (one study[66]); EFW or AC for gestational age <10th centile plus ultrasound parameters (two studies[70,72]), others (four studies[40,56,59,69]); and unreported (one study[48]). Overall, according to traditional definitions for FGR and SGA outlined in the Methods section, 31 studies included fetuses/infants considered SGA[28–33,35–37,39,41,42,44–47,50,51,53,54,57,59,61–64,66,68,71,73,74], 12 included fetuses/infants considered growth-restricted[27,34,49,52,56,58,60,65,67,69,70,72], four included both SGA and fetuses/infants considered growth-restricted[38,40,43,55] and in one it was unknown[48]. There were several categories of reference group: birthweight for gestational age between the 10th-90th centiles (19 studies[30,32,33,35,38–40,45,50,54,55,61,62,64,68,70,71,73,74]); birthweight ≥10th centile (eight studies[34,44,47,49,56–58,63]); other intervals of birthweight centiles (seven studies[31,42,43,46,51,52,67]); "uncomplicated/normal/healthy pregnancies or healthy/AGA newborns" (nine studies[27–29,37,41,48,53,66,72]); birthweight Z-score between −1 and +1 SD (one study[59]); and unreported (four studies[36,60,65,69]).

Twenty-six studies used an untargeted metabolomic approach[27–31,34–37,39,41,45–47,49,52–54,59,60,66,69–72,74], 18 used a targeted approach[32,33,38,40,42,48,50,51,55,58,61–65,67,68,73], and four used both approaches[43,44,56,57]. The analytical platforms used for metabolite detection included liquid chromatography (ultra/high/ultra-high performance) coupled to mass spectrometry (LC-MS) in 24 studies[27,29,30,33,38,39,45,48–53,55,59,61–65,68–70,74], nuclear magnetic resonance (NMR) spectroscopy in 11 studies[28,31,34–36,43,46,54,56,60,67], gas chromatography coupled to mass spectrometry (GC–MS) in seven studies[32,37,40,47,66,71,72], flow injection analysis (FIA) in one study[42], LC-MS and NMR in two studies[44,57], LC-MS and FIA in two studies[58,73], and LC-MS and GC-MS in one study[41].

Multivariate approaches that were used to analyse metabolites individually, as well as the relationships among the individual metabolites, included multivariate linear regression models (15 studies[33,36,37,41,42,45,49,50,53,55,56,62,63,67,72]), partial least square discriminant analysis (PLS-DA) or orthogonal PLS-DA (seven studies[29,43,44,54,57,69,71]), principal component analysis (PCA) (three studies[27,30,65]), and both PLS-DA/orthogonal PLS-DA and PCA (12 studies[28,31,34,35,39,46,58–60,66,73,74]). Software for metabolic pathway analysis was used in 12 studies[28,39,44,46,48,57,66,69–71,73,74].

The risk of bias in each included study is summarised in Supplementary Fig. 1. No study was judged to be at low risk of bias for all eight domains. Only 13 studies (27%) fulfilled at least six of the eight criteria for low risk of bias[29,34,43,45,49,53,55,58,60,69,71,72,74]. Eleven studies were deemed to be at low risk of bias for five domains[31,33,44,46,47,50,57,65,67,70,73], whereas the remaining 24 studies (50%) had four or more methodological flaws[27,28,30,32,35–42,48,51,52,54,56,59,61–64,66,68]. The most common deficiencies were related to unblinded interpretation of metabolomics results to fetal growth status of participants, overfitting in the analyses, and lack of reporting on handling of specimens and pre-analytical procedures.

Among the 13 studies that met at least six of the eight criteria for low risk of bias, 11[29,34,43,49,53,55,58,60,71,72,74] provided data for metabolomic profiles and pathway analyses in different biological samples. The remaining two studies did not provide data to these analyses[45,69].

Overall, a total of 825 non-duplicated metabolites were significantly altered across the 48 included studies, of which 46% were up- and 54% down-regulated. Eighty significantly altered metabolites were reported in more than one study (fatty acyls, 35%; amino acids, 31%; glycerophospholipids, 21%; others, 13%), 29 in more than two studies (amino acids, 52%; fatty acyls, 24%; glycerophospholipids, 21%; others, 3%), and 20 in more than three studies (amino acids, 60%; glycerophospholipids, 20%; fatty acyls, 20%).

## Metabolomic profiles in maternal plasma or serum

Nineteen studies assessed metabolomic profiles in maternal plasma (11 studies[29,31,40,43,48,50,53,56,65,73,74]) and serum (eight studies[32,42,49,52,54,64,70,72]): three only ≤20 weeks' gestation[29,53,64], two between 24 and 28 weeks' gestation[52,73], three in the third trimester[65,70,74], two collected serial samples in the first, second and third trimester of pregnancy[49,50], and nine collected the samples within 24 h before birth, or during or after birth[31,32,40,42,43,48,54,56,72] (Table 1).

Six studies (one at ≤20 weeks' gestation[64], one at 24−28 weeks' gestation[73] and four in the peripartum period[31,32,42,56]) did not identify significantly up- or down-regulated metabolites. In the remaining 13 studies (three at ≤20 weeks' gestation[29,50,53], five at >20 weeks' gestation[49,52,65,70,74], and five in the peripartum period[40,43,48,54,72]), a total of 156 non-duplicated metabolites had significantly different concentrations between the FGR/SGA and the corresponding reference groups (103 up-regulated and 53 down-regulated). Eight of the 156 metabolites were significantly up- or down-regulated in more than one study (pregnanediol-3-glucuronide in studies at ≤20 weeks' gestation; malic acid in studies at >20 weeks' gestation and in the peripartum period; and alanine, isoleucine, lysine, serine, phenylalanine, and 4-aminobutyric acid in studies in the peripartum period) and only one (alanine in studies in the peripartum period) in more than two studies (Table 2).

## Metabolomic profiles in maternal plasma or serum at ≤ 20 weeks' gestation

Five studies[29,49,50,53,64], assessed metabolomic profiles in plasma or serum of pregnant women at ≤20 weeks' gestation (Table 1). Up-regulated metabolites in FGR/SGA pregnancies that were reported in individual studies included cervonyl carnitine and sphingolipids-related metabolites[29], plasmalogen[49], eicosanoids related to hydroxyeicosatetraenoic and dihydroxyeicosatrienoic acids[50], and glycerophospholipids (mainly phosphatidylserines, phosphatidylethanolamines and phosphatidylcholines), sphingolipids, glycerolipids, and fatty acyls[53]. Two studies[29,49] reported that steroids-related metabolites were usually down-regulated in SGA pregnancies. One study[64] did not find any significantly altered metabolites between women who subsequently delivered SGA infants and controls at 13 weeks' gestation.

The total number of metabolites significantly different between the FGR/SGA and reference groups was 52 (45 up- and seven down-regulated), of which only one (pregnanediol-3-glucuronide) was significantly and consistently down-regulated in more than one study (Table 2), primarily likely due to differences in analytical platforms and focus on different metabolite classes of each study from the description provided.

## Metabolomic profiles in maternal plasma or serum at > 20 weeks' gestation and in the peripartum period

Sixteen studies evaluated metabolomic profiles at >20 weeks' gestation (n = 7[49,50,52,65,70,73,74]) or in the peripartum period (n = 9[31,32,40,42,43,48,54,56,72]) (Table 1). Most studies evaluating metabolomic profiles at 24−28 weeks' gestation and in the third trimester reported few metabolites significantly altered[50,52,65,73,74]. Only one study[70] identified 13 significantly altered metabolites in the third trimester (nine up- and four down-regulated). Among the nine studies that evaluated metabolomic profiles in the peripartum period, four[31,32,42,56] did not find any significantly altered metabolites and three[40,43,54] reported only a few altered metabolites. In the remaining two studies, one[48] reported 18 significantly altered metabolites (15 up-regulated, mainly amino acids, and three down-regulated) and the other[72] reported 31 altered metabolites (13 up-regulated, mostly derivatives of amino acids, keto acids and carboxylic acids, and 18 down-regulated, mostly unsaturated and saturated fatty acids and organic compounds).

**Table 2 | Significantly up-regulated and down-regulated metabolites that were reported in >1 study in maternal biological samples**

| Metabolite | No. of studies | Up-regulated, No. of studies | Down-regulated, No. of studies |
|---|---|---|---|
| Maternal plasma/serum | | | |
| ≤20 weeks' gestation (5 studies[29,49,50,53,64]) | | | |
| Consistent trend[a] | | | |
| Pregnanediol-3-glucuronide | 2 | 0 | 2[29,49] |
| Inconsistent trend | | | |
| None | --- | --- | --- |
| >20 weeks' gestation/peripartum period (16 studies[31,32,40,42,43,48–50,52,54,56,65,70,72–74]) | | | |
| Consistent trend[a] | | | |
| Isoleucine | 2 | 2[48,54] | 0 |
| Lysine | 2 | 2[48,72] | 0 |
| Serine | 2 | 2[48,72] | 0 |
| 4-aminobutyric acid | 2 | 2[48,72] | 0 |
| Malic acid | 2 | 0 | 2[70,72] |
| Inconsistent trend | | | |
| Alanine | 4 | 3[48,54,72] | 1[43] |
| Phenylalanine | 2 | 1[48] | 1[54] |
| Maternal hair | | | |
| ≤20 weeks' gestation (no studies) | | | |
| None | --- | --- | --- |
| >20 weeks' gestation/peripartum period (2 studies[37,41]) | | | |
| Consistent trend[a] | | | |
| Margaric acid | 2 | 2[37,41] | 0 |
| Myristic acid | 2 | 2[37,41] | 0 |
| Inconsistent trend | | | |
| None | --- | --- | --- |
| Maternal urine | | | |
| ≤20 weeks' gestation (3 studies[36,47,53]) | | | |
| None | --- | --- | --- |
| >20 weeks' gestation/peripartum (1 study[47]) | | | |
| None | --- | --- | --- |

[a]The trend of one metabolite was considered consistent if it showed the same direction of change in all studies.

The total number of metabolites significantly different between the FGR/SGA and reference groups was 109 (44 at >20 weeks' gestation [27 up- and 17 down-regulated], and 65 in the peripartum period [32 up- and 33 down-regulated]). Overall, seven metabolites were significantly altered in more than one study, of which five showed a consistent trend (isoleucine, lysine, serine and 4-aminobutyric acid, up-regulated in all studies; and malic acid, down-regulated in all studies) and two an inconsistent trend (alanine, up-regulated in three studies and down-regulated in one; and phenylalanine, up-regulated in one study and down-regulated in one study) (Table 2).

**Metabolomic profiles in maternal hair**
Two studies investigated the metabolomic profiles of maternal hair samples in SGA cases, one at 26-28 weeks' gestation[37] and the other in the second and third trimesters of pregnancy[41] (Table 1). Overall, a total of 33 non-duplicated metabolites had significantly different concentrations between the SGA and reference groups (11 up- and 22 down-regulated), of which only two were significantly altered in more than one study (margaric acid and myristic acid, both up-regulated in the two studies) (Table 2). In one study[37], most of the 32 significantly altered metabolites reported were amino acids, amino acid derivatives

**Table 3 | Significantly up-regulated and down-regulated metabolites that were reported in >1 study in placenta samples**

| Metabolite | No. of studies | Up-regulated, No. of studies | Down-regulated, No. of studies |
|---|---|---|---|
| Consistent trend[a] | | | |
| Glycine | 2 | 0 | 2[57,71] |
| Glycerophosphocholine | 2 | 2[60,71] | 0 |
| Lactic acid | 2 | 2[60,71] | 0 |
| Inconsistent trend | | | |
| Taurine | 3 | 1[60] | 2[57,71] |
| Glutamine | 2 | 1[60] | 1[57] |
| Asparagine | 2 | 1[71] | 1[57] |
| Aspartic acid | 2 | 1[71] | 1[57] |
| Tyrosine | 2 | 1[71] | 1[57] |
| Isoleucine | 2 | 1[71] | 1[57] |
| Leucine | 2 | 1[72] | 1[57] |

[a]The trend of one metabolite was considered consistent if it showed the same direction of change in all studies.

and fatty acids. The other study[41] reported that three metabolites (all long-chain fatty acids) were significantly up-regulated in the second trimester and none in the third trimester.

**Metabolomic profiles in maternal urine**
Three studies evaluated metabolomic profiles of FGR/SGA in maternal urine, two at ≤20 weeks' gestation[36,53] and one at 10 and 26 weeks' gestation[47] (Table 1). A total of 20 non-duplicated metabolites had significantly different concentrations between the FGR/SGA and reference groups (nine up- and 11 down-regulated). None of these metabolites were significantly altered in more than one study (Table 2). One[47] of the three studies found only up-regulated metabolites and two[36,53] found only down-regulated metabolites. Lower levels of metabolites involved in nutrient transport and detoxification pathways in women with SGA pregnancies were reported in one study[53]. The remaining two studies[36,47] did not identify any perturbed pathways.

**Metabolomic profiles in maternal faeces**
Metabolomic profiles of FGR/SGA in maternal faeces were examined in one study[70], which reported significant differences in the concentrations of 23 metabolites (seven up- and 16 down-regulated) in women with FGR pregnancies compared to women with AGA pregnancies in the third trimester (Table 1). Pathway analysis showed that lipid, amino acid, sphingolipid, fatty acid, and steroid hormone metabolism was enriched in the FGR group.

**Metabolomic profiles in amniotic fluid**
Only one study[66] evaluated the metabolomic profiles of FGR/SGA pregnancies in amniotic fluid at a mean gestational age of 30 weeks (Table 1). A total of 47 differentially expressed metabolites were identified of which 23 were up-regulated (mainly metabolites involved in glucose metabolism such as malic acid, glycolic acid, maleic acid, and D-glycerate) and 24 down-regulated (mainly amino acids such as glutamate, phenylalanine, valine and leucine).

**Metabolomic profiles in placenta**
Five studies assessed metabolomic profiles of FGR/SGA in placental samples[27,57,60,71,72], of which one[27] did not provide clear data on altered metabolites. Overall, a total of 217 non-duplicated metabolites were reported to be significantly altered across studies (38 up- and 179 down-regulated), of which just 10, mostly amino acids, were significantly altered in more than one study (Table 3) and one (taurine) was significantly altered in more than two studies. Importantly, only

**Table 4 | Significantly up-regulated and down-regulated metabolites that were reported in >1 study in umbilical cord blood samples**

| Metabolite | No. of studies | Up-regulated, No. of studies | Down-regulated, No. of studies |
|---|---|---|---|
| **Consistent trend[a]** | | | |
| LysoPC (16:1) | 6 | 0 | 6[29,42,58,59,64,73] |
| PC (36:3) | 3 | 0 | 3[58,59,64] |
| Leucine | 3 | 3[34,39,54] | 0 |
| Choline | 3 | 0 | 3[31,34,44] |
| Triglyceride | 3 | 3[34,43,56] | 0 |
| Glutamic acid | 2 | 2[30,39] | 0 |
| Trans-4-hydroxyproline | 2 | 2[58,72] | 0 |
| LysoPC (14:0) | 2 | 0 | 2[42,59] |
| LysoPC (16:0) | 2 | 0 | 2[58,73] |
| LysoPC (18:0) | 2 | 0 | 2[58,73] |
| LysoPC (20:4) | 2 | 0 | 2[58,73] |
| PC (36:1) | 2 | 0 | 2[58,59] |
| PC (36:4) | 2 | 0 | 2[44,59] |
| PC (38:4) | 2 | 0 | 2[44,59] |
| PC (40:4) | 2 | 0 | 2[44,59] |
| Decanoyl carnitine | 2 | 2[59,73] | 0 |
| Dodecanoid acid | 2 | 2[32,40] | 0 |
| 2-aminoadipic acid | 2 | 2[58,72] | 0 |
| Stearic acid | 2 | 0 | 2[32,72] |
| Gamma-linolenic acid | 2 | 0 | 2[32,72] |
| Eicosatrienoic acid | 2 | 0 | 2[32,72] |
| Arachidonic acid | 2 | 0 | 2[32,72] |
| Cholesterol HDL | 2 | 0 | 2[56,67] |
| Glucose | 2 | 0 | 2[31,34] |
| **Inconsistent trend** | | | |
| Phenylalanine | 6 | 4[30,31,39,72] | 2[34,54] |
| LysoPC (18:1) | 5 | 1[44] | 4[42,58,59,73] |
| Alanine | 5 | 3[39,54,58] | 2[31,34] |
| Valine | 4 | 3[30,39,54] | 1[34] |
| Isoleucine | 4 | 3[30,39,54] | 1[56] |
| Glutamine | 4 | 2[34,58] | 2[31,34] |
| LysoPC (18:2) | 4 | 1[44] | 3[29,58,73] |
| LysoPC (20:3) | 4 | 1[44] | 3[58,59,73] |
| Carnitine | 4 | 3[39,58,74] | 1[44] |
| Tryptophan | 3 | 1[30] | 2[54,58] |
| Proline | 3 | 2[30,58] | 1[31] |
| Tyrosine | 3 | 2[39,58] | 1[34] |
| Histidine | 3 | 1[30] | 2[39,67] |
| PC (38:3) | 3 | 1[44] | 2[58,59] |
| Methionine | 2 | 1[30] | 1[39] |
| Arginine | 2 | 1[30] | 1[39] |
| PC (24:0) | 2 | 1[44] | 1[58] |
| PC (32:0) | 2 | 1[44] | 1[58] |
| Acetyl carnitine | 2 | 1[58] | 1[44] |
| Butiryl carnitine | 2 | 1[58] | 1[44] |
| Hexacosanedioic acid | 2 | 1[59] | 1[29] |
| Caffeine | 2 | 1[30] | 1[59] |

*HDL* high-density lipoprotein, *PC* phosphatidylcholine.
[a]The trend of one metabolite was considered consistent if it showed the same direction of change in all studies.

three metabolites from the initial 217, had a consistent trend: glycerophosphocholine and lactic acid, up-regulated in two studies[60,71]; and glycine, down-regulated in two studies[57,71]. The remaining seven significantly altered metabolites (all amino acids) had an inconsistent trend across the included studies (Table 3). Pathway analysis from one study[57] revealed abnormalities that were consistent with fetal hepatic dysfunction in suspected FGR. Another study[71] reported that metabolic pathways related to the hypoxia response and amino-acid uptake and metabolism were associated with SGA.

### Metabolomic profiles in umbilical cord blood
Among the 21 studies that assessed metabolomic profiles in umbilical cord blood, 20 reported significant differences in metabolite concentrations between the FGR/SGA and reference groups. A total of 308 non-duplicated metabolites were significantly altered (155 up- and 153 down-regulated), of which 45 metabolites were significantly altered in more than one study and 18 in more than two studies (Table 4).

The amino acids phenylalanine, alanine, valine, isoleucine, and glutamine, four lysophosphatidylcholines (16:1, 18:1, 18:2, 20:3), and carnitine were the most reported altered metabolites. Of the 46 significantly altered metabolites in more than one study, 24 showed a consistent trend across studies: 17 were down-regulated in all studies (five lysophosphatidylcholines, five phosphatidylcholines, three fatty acids, gamma-linolenic acid, choline, cholesterol, and glucose) and seven were up-regulated in all studies (four amino acids, decanoyl carnitine, dodecanoid acid, and triglyceride). The remaining 22 metabolites showed an inconsistent trend (Table 4).

### Metabolomic profiles in newborn dried blood spots
Seven studies, all using a targeted approach, evaluated metabolomic profiles of SGA in dried blood spots taken from a heel prick between 12 hours and 8 days after birth for newborn metabolic screening[33,38,51,55,61,62,68]. In general, metabolites associated with acylcarnitine were upregulated in most studies. Only one study reported that most acylcarnitines assessed were down-regulated[68].

A total of 112 non-duplicated metabolites had significantly different concentrations between SGA and AGA neonates (80 up- and 32 down-regulated), of which 31 were significantly altered in more than one study and 12 in more than two studies (Table 5). Of the 31 significantly altered metabolites in more than one study, 18 showed a consistent trend across studies: 14 were up-regulated in all studies (10 acylcarnitines and four amino acids) and four were down-regulated in all studies (three amino acids and one acylcarnitine). The remaining 13 significantly altered metabolites in more than one study had an inconsistent trend (nine acylcarnitines and four amino acids).

### Metabolomic profiles in newborn urine
Three studies evaluated metabolomic profiles of FGR in newborn urine[28,35,69]. All samples were taken within 48 h of birth. A total of 76 non-duplicated metabolites were significantly altered across studies (31 up- and 45 down-regulated) of which three were significantly and consistently altered in more than one study (myo-inositol, creatinine and creatine, up-regulated in all studies) (Table 5). No metabolite was significantly altered in more than two studies. One study[28] reported three metabolic pathways associated with FGR (one involved in the metabolism of arginine and proline, one associated with the urea cycle and the third correlated with the metabolism of glycine, serine and threonine) and another[69] reported metabolic pathways related to tryptophan and histidine metabolism and aminoacyl-tRNA and steroid hormone biosynthesis.

### Metabolomic profiles in breast milk
Metabolomic profiles of SGA in breast milk were assessed in one study[46], which reported significantly different concentrations of seven metabolites (five up- and two down-regulated) in milk/colostrum on

**Table 5 | Significantly up-regulated and down-regulated metabolites that were reported in >1 study in neonatal samples**

| Metabolite | No. of studies | Up-regulated, No. of studies | Down-regulated, No. of studies |
|---|---|---|---|
| **Newborn dried blood spot** | | | |
| Consistent trend[a] | | | |
| Free carnitine | 5 | 5[33,51,55,61,62] | 0 |
| Butyryl carnitine | 3 | 3[51,61,62] | 0 |
| Acetyl carnitine | 3 | 3[33,51,62] | 0 |
| Decenoyl carnitine | 3 | 3[51,61,62] | 0 |
| Propionyl carnitine | 3 | 0 | 3[51,61,62] |
| Proline | 3 | 3[55,61,62] | 0 |
| Phenylalanine | 2 | 2[51,62] | 0 |
| Leucine | 2 | 2[51,55] | 0 |
| Glycine | 2 | 2[51,62] | 0 |
| Tyrosine | 2 | 0 | 2[33,38] |
| Valine | 2 | 0 | 2[55,62] |
| Arginine | 2 | 0 | 2[62,68] |
| Octadecadienyl carnitine | 2 | 2[33,62] | 0 |
| Isovaleryl carnitine | 2 | 2[38,51] | 0 |
| Tetradecanoyl carnitine | 2 | 2[51,62] | 0 |
| Dodecanoyl carnitine | 2 | 2[61,62] | 0 |
| Octadecanoyl carnitine | 2 | 2[61,62] | 0 |
| Octadecenoyl carnitine | 2 | 2[61,62] | 0 |
| Inconsistent trend | | | |
| Alanine | 6 | 5[33,51,55,61,62] | 1[68] |
| Ornithine | 6 | 4[38,55,61,62] | 2[38,68] |
| Methionine | 4 | 3[51,55,62] | 1[38] |
| Citrulline | 3 | 2[61,62] | 1[68] |
| Octanoyl carnitine | 3 | 2[61,62] | 1[68] |
| Decanoyl carnitine | 3 | 2[61,62] | 1[68] |
| Dodecenoyl carnitine | 2 | 1[61] | 1[68] |
| Tetradecadienoyl carnitine | 2 | 1[61] | 1[68] |
| Hexadecenoyl carnitine | 2 | 1[61] | 1[62] |
| OH-Octadecenoyl carnitine | 2 | 1[61] | 1[62] |
| Malonyl carnitine | 2 | 1[62] | 1[68] |
| Hexanoyl carnitine | 2 | 1[62] | 1[68] |
| Acyl carnitine | 2 | 1[62] | 1[68] |
| **Newborn urine** | | | |
| Consistent trend[a] | | | |
| Myo-inositol | 2 | 2[28,35] | 0 |
| Creatinine | 2 | 2[28,35] | 0 |
| Creatine | 2 | 2[28,35] | 0 |
| Inconsistent trend | | | |
| None | --- | --- | --- |

[a]The trend of one metabolite was considered consistent if it showed the same direction of change in all studies.

the third to fourth day postpartum between mothers of SGA infants and controls (Table 1).

## Analysis of metabolic pathways

Despite several metabolic pathways being significantly enriched in unadjusted analyses, only four metabolic pathways were found to be significantly enriched in adjusted analyses (FDR < 0.05): one in umbilical cord blood (biosynthesis of unsaturated fatty acids with an FDR *p*

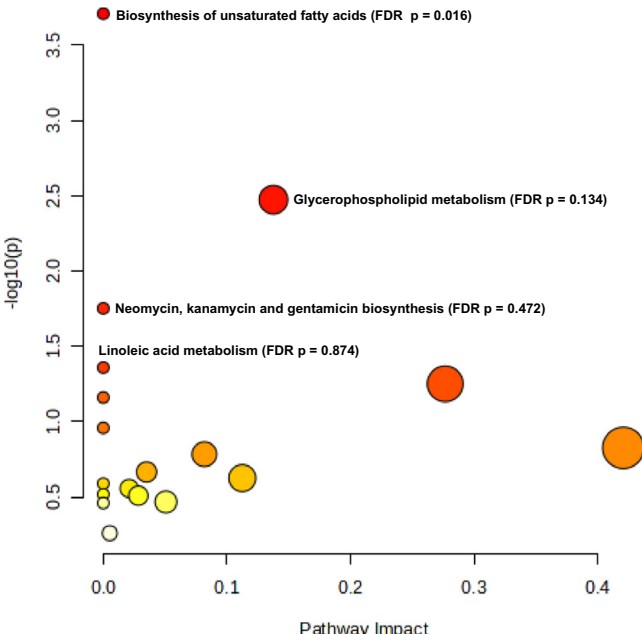

**Fig. 2 | Pathway analysis for significantly and consistently up- and down-regulated metabolites (*N* = 24) that were reported in more than one study in umbilical cord blood samples.** The metabolome view shows all matched pathways according to the p values from the pathway enrichment analysis and pathway impact values from the pathway topology analysis. Each circle in the figure represents a metabolic pathway. The colour of the circle indicates the significance level (Raw p) in the enrichment analysis; darker colour (redder) indicates greater significance. The size of the circle reflects the pathway impact value in the topology analysis, such that the larger the circle, the larger the impact value. Only the biosynthesis of unsaturated fatty acids was found to be significantly enriched in adjusted analyses (false discovery rate *p* value < 0.05). FDR false discovery rate. Source data are provided as a Source Data file.

value of 0.016 and an impact value of 0.0) (Fig. 2 and Supplementary Table 1) and three in newborn dried blood spots (phenylalanine, tyrosine and tryptophan biosynthesis; valine, leucine and isoleucine biosynthesis; and phenylalanine metabolism, with FDR *p* values of 0.014, 0.021, and 0.021, respectively, and impact values of 1.0, 0.0, and 0.36, respectively) (Fig. 3 and Supplementary Table 2). Dried blood spots were taken for newborn screening of inborn metabolic diseases, although studies reported that newborns with genetic metabolic diseases were excluded from analyses.

There were no significantly enriched metabolic pathways (FDR ≥ 0.05) in maternal plasma/serum at >20 weeks' gestation or in the peripartum period (Supplementary Fig. 2 and Supplementary Table 3) and in placenta (Supplementary Fig. 3 and Supplementary Table 4). Pathway analyses in maternal plasma/serum at ≤20 weeks' gestation, maternal hair, and newborn urine could not be performed because of there were only few significantly and consistently up- or down-regulated metabolites in more than one study in such biological samples.

## Discussion
### Principal findings
A total of 825 non-duplicated metabolites were significantly altered (46% up- and 54% down-regulated) across the 48 studies included in this systematic review using 10 different human biological samples. Only 56 metabolites (17 amino acids, 12 acylcarnitines, 11 glycerophosphocholines, six fatty acids, two hydroxy acids, and eight other metabolites) were reported to be significantly up- or down-regulated in more than one study with a consistent direction of the association, i.e. up- or down-regulated in all studies reporting that metabolite. Only pregnanediol-3-glucuronide was reported consistently down-regulated

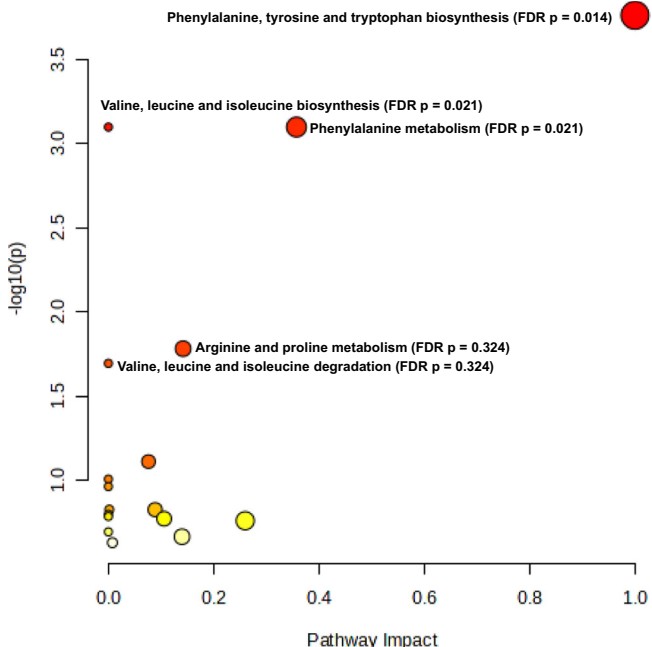

**Fig. 3 | Pathway analysis for significantly and consistently up- and down-regulated metabolites (N = 18) that were reported in more than one study in newborn dried blood spot samples.** The metabolome view shows all matched pathways according to the *p* values from the pathway enrichment analysis and pathway impact values from the pathway topology analysis. Each circle in the figure represents a metabolic pathway. The colour of the circle indicates the significance level (Raw p) in the enrichment analysis; darker colour (redder) indicates greater significance. The size of the circle reflects the pathway impact value in the topology analysis, such that the larger the circle, the larger the impact value. Three metabolic pathways were found to be significantly enriched in adjusted analyses (false discovery rate *p* value < 0.05): phenylalanine, tyrosine and tryptophan biosynthesis; valine, leucine and isoleucine biosynthesis; and phenylalanine metabolism. FDR false discovery rate. Source data are provided as a Source Data file.

in maternal samples at ≤20 weeks' gestation, a period very relevant for the potential prediction of FGR/SGA. The remaining 55 metabolites were reported in maternal plasma at >20 weeks' gestation or in the peripartum period (n = 5); maternal hair at >20 weeks' gestation (n = 2); placenta (n = 3); umbilical cord blood (n = 24); newborn dried blood spots (n = 18), and newborn urine (n = 3) (Supplementary Box 1).

Three amino acid metabolism-related pathways and one related with lipid metabolism were significantly associated with FGR and/or SGA: biosynthesis of unsaturated fatty acids in umbilical cord blood, and phenylalanine, tyrosine and tryptophan biosynthesis, valine, leucine and isoleucine (branched chain amino acids, BCAAs) biosynthesis, and phenylalanine metabolism in newborn dried blood spot. Among these pathways, phenylalanine, tyrosine and tryptophan biosynthesis and phenylalanine metabolism had the highest impact values (1.0 and 0.36, respectively). Significantly enriched metabolic pathways were not identified in the remaining biological samples. Observationally, however, across blood samples, those taken from mothers (at >20 weeks' gestation), umbilical cord blood, and newborn dried bloodspots, showed perturbation of BCAA metabolism (i.e., the concentrations of isoleucine/valine/leucine). This may be particularly pertinent since BCAAs are essential amino acids and can only be, in humans, derived from the diet. BCAAs are also key for stimulating protein biosynthesis and tissue development[75–78]. Moreover, growth-restricted compared to AGA fetuses have lower plasma concentrations of BCAAs in the umbilical artery and vein[79]. Hence, one can speculate that common perturbations of BCAA metabolism in blood samples identified in this systematic review may contribute to, or reflect, impaired growth of the fetus.

## Comparison with existing literature

Only one previous systematic review, including 21 studies, has evaluated metabolomic profiles in FGR[80]. Eighteen metabolites were identified that were significantly altered (unreported definition of statistical significance) in more than two studies (nine in neonatal studies [cord blood and newborn dried blood spot] and nine in maternal studies [maternal serum/plasma, urine, and hair, placenta and milk]) of which alanine, valine and isoleucine were reported in both maternal and neonatal studies. Other metabolites that were significantly altered in more than two studies included citrate and glycine in maternal studies, and proline, phenylalanine, and glutamine in neonatal studies. The most significantly enriched metabolic pathways with relatively high impact values were glutathione metabolism in maternal studies, glyoxylate and dicarboxylate metabolism and alanine, aspartate, and glutamate metabolism in neonatal studies, and arginine biosynthesis and arginine and proline metabolism in both maternal and neonatal studies. Conversely, our review did not identify any of these metabolic pathways as significantly enriched. Such a discrepancy could be explained by the smaller number of studies included by Yao et al.[80]; stratification of studies, using different biological samples, into only maternal and neonatal studies; and the inclusion of all significantly altered metabolites in pathway analysis without considering the frequency and consistency of the association.

## Strengths and limitations

The major strengths of the present review are: (1) the rigorous methodology used and the complete adherence to the MOOSE guidelines including following the defined format for summarising the evidence[22]; (2) the inclusion of the largest number of mostly recent studies reported from different populations throughout the world; (3) the inclusion of studies assessing the association between FGR/SGA and metabolomics in 10 different biological samples; (4) the stringent criteria used for including metabolites in pathway analyses; and (5) separating those studies that examined metabolomic profiles in umbilical cord blood from those that used newborn dried blood spot and urine samples.

Despite these efforts, there are specific areas of concern with these data. First, there was substantial heterogeneity among studies in terms of design; sample size; participant characteristics; case, control and outcome definitions; timing of sampling; sample collection and preparation; data acquisition and processing; metabolomic methodologies used; and analytical and statistical approaches used, which limits the possibility of a summary statement. Such large heterogeneity could explain the inconsistent trends and conflicting patterns of metabolites significantly associated with FGR/SGA. Considerably greater efforts are needed to improve the standardised reporting of metabolomic studies following recent suggestions[81].

Second, a major source of variation across studies was: (1) the use of very different definitions of FGR and SGA (that are often wrongly used interchangeably in the literature) and categories of reference groups, and (2) the failure to recognise the syndromic nature of these two anthropometric and clinical entities that have multiple interrelated aetiologies and risk factors. These limitations considerably undermine both the internal and external validity of studies. Hence, it is possible that the conflicting results just represent the role of the different aetiological factors associated with sub-groups of FGR/SGA that constitute the underlying risk profile of the samples selected.

Third, several studies only reported the significantly altered metabolites and no information was provided on the metabolites with non-significant differences in concentrations between the FGR/SGA and reference groups (selective non-reporting bias). There is also the risk of publication bias of research findings, depending on the nature and direction of the results, especially in studies exploring predictive biomarkers.

Fourth, although NMR has a significantly lower sensitivity and detects a much smaller number of metabolites than MS-based methods, both approaches are complementary. However, most studies included in our review only used one analytical method and made no comparisons across platforms. This may have resulted in metabolites being identified on one platform and not another, resulting in less consistency across studies.

Finally, there are two highly relevant conceptual issues for interpreting this literature: (1) the reliability of the results of a systematic review is limited by the methodological quality of the studies included. In our review, only just over a quarter of included studies met at least six of the eight criteria for low risk of bias. In addition, most included studies were case-control and cross-sectional, thus limiting the power to verify causal relationships between altered metabolites and FGR/SGA, which means that reverse causality should be carefully examined since metabolites could be the result of FGR/SGA rather than a cause. This limitation is key to maternal samples collected close to birth, as well as umbilical cord blood and newborn samples. Importantly, none of the included studies assessed neuro-developmental outcomes and only one study[58] evaluated postnatal growth patterns up to the age of 12 months. (2) The use of pathway analysis methods has intrinsic limitations, such as arbitrary criteria for defining pathways and *p* value cut-offs for selecting significant metabolites; input data and parameters used; changes in the background set; reliability of compound identification, and database updates, among others. Moreover, the statistical techniques used in pathway analysis consider only the number of statistically significant metabolites without taking into account the measured fold changes and trend consistency.

It might also be said that the results of pathway models are self-fulfilling: if a metabolomic study, especially if targeted, identifies molecules of related families (such as amino acids, fatty acids, and markers of glucose metabolism) associated with a phenotype, the pathway models will inevitably report that amino acid, lipid, and carbohydrate metabolic pathways are affected by the outcome of interest. Such a summary description of the underlying metabolic processes involved in complex syndromes is not necessarily useful for the identification of therapeutic strategies at the molecular level.

## Clinical and research implications

In conclusion, our systematic review identified a number of altered metabolites and metabolic pathways that were associated with FGR and/or SGA. Some of these metabolites appear promising and may provide new insights for understanding the pathophysiology of these syndromes and the development of new therapeutic targets. Promising metabolites include lysophosphatidylcholine 16:1, phosphatidylcholine 36:3, leucine, choline, and triglyceride in umbilical cord blood samples, free carnitine, butyryl carnitine, acetyl carnitine, decenoyl carnitine, propionyl carnitine, and proline in newborn dried blood spot, and pregnanediol-3-glucuronide in maternal early-pregnancy samples. Well-designed and phenotyped studies with a large number of FGR/SGA cases to allow for stratification according to aetiology, especially longitudinal cohort metabolomics in plasma or serum of pregnant women and clinical intervention metabolomics studies, should be carried out to explore novel biomarkers of FGR/SGA and determine target metabolic pathways for prevention and treatment. Integrating metabolomic and other omic data would seem to be the next step to better elucidate networks of molecular mechanisms in FGR/SGA.

## Methods

As a systematic review, our study did not involve direct participation of human subjects and focused solely on previously published and publicly available data. It did not require institutional review board approval for this reason. The ethical principles governing this study adhere to the established guidelines for systematic reviews and meta-analyses. This systematic review was registered with PROSPERO (CRD42021275753) on September 23, 2021 (https://www.crd.york.ac.uk/prospero/display_record.php?RecordID=275753) and reported according to the MOOSE guidelines for meta-analyses of observational studies[82]. Two authors (AC-A and MR) initially examined the relevant literature; AC-A and JV independently reviewed studies for inclusion, assessed their risk of bias, and extracted data. Disagreements were resolved through consensus.

### Literature search

We searched MEDLINE, EMBASE, LILACS, CINAHL, Scopus, Web of Science, and the Cochrane Central Register of Controlled Trials (all from 1998, the year that the term metabolomics was introduced, to 31 December 2023) using a combination of keywords and text words related to *metabolomics* ("metabolomic", "metabonomic", "metabolome", "metabolite", "lipidomic", "oxylipins", "lipid mediators", "proton nuclear magnetic resonance", "liquid chromatography", "gas chromatography", "high-performance liquid chromatography", "ultra-performance liquid chromatography") and *FGR* and *SGA* ("fetal growth restriction", "fetal growth retardation", "impaired fetal growth", "intrauterine growth restriction", "intrauterine growth retardation", "small for gestational age", "small for date", "small for gestation"). Google Scholar, proceedings of congresses and scientific meetings on obstetrics, maternal-fetal medicine and omics technologies, reference lists of identified studies, previously published systematic reviews, and review articles were also searched. We did not use any language restrictions. The initial search was performed from 1 June 2023 to 15 June 2023. Searches were re-run on a monthly basis until 2 January 2024.

### Eligibility criteria

We included observational (cohort, case-control and cross-sectional) studies that reported on associations between metabolites measured using any metabolomic technology in tissues and biofluids of (a) women with a singleton pregnancy or (b) singleton newborns (within the first 7 days of life) and FGR or SGA diagnosed by criteria defined by the authors. Acceptable definitions for FGR included EFW or AC or birthweight below the 10th, 5th, or 3rd centiles for gestational age/sex (as reported by the authors) plus indicators of fetal and placental health such as amniotic fluid volume, biophysical profile, maternal and fetal Doppler velocimetry, biomarkers, and placental pathology, among others. Acceptable definitions for SGA included birthweight below the 10th, 5th, or 3rd centiles or less than two standard deviations below the mean for gestational age/sex regardless of birthweight reference or standard used. Analysed samples included maternal blood, urine, faeces and hair, amniotic fluid, placenta, breast milk, umbilical cord blood, and neonatal blood and urine. Studies not using a metabolomics technology, animal studies, studies including multiple pregnancies or neonates that did not report singleton data separately, conference abstracts, case reports, letters, editorials, and reviews were excluded from the review.

### Assessment of risk of bias

The risk of bias in each included study was assessed using a modified version of QUADOMICS[83], an adaptation of the Quality Assessment of Diagnostic Accuracy Studies (QUADAS)[84] tool for studies using omic technologies. A total of eight domains were assessed. Each domain was judged as having a "low," "high," or "unclear" risk of bias. The domains evaluated and how they were interpreted were as follows:

1. Selection of participants – "low risk of bias": all participants were selected from the same population and during the same time period; "high risk of bias": all participants were not selected from the same population and/or were not selected during the same time period.

2. Description of selection criteria – "low risk of bias": if detailed information on inclusion/exclusion criteria and sources of samples was reported; "high risk of bias": if this information was not reported.

3. Description of procedures and timing of biological sample collection with respect to clinical factors – "low risk of bias": the study report included an analysis of potential factors affecting the metabolite profile, and a procedure to control biases that they might induce; "high risk of bias": if this information was not reported.

4. Reporting of handling of specimens and pre-analytical procedures and if they were similar for the whole sample – "low risk of bias": the study reported that the whole set of samples underwent the same pre-analytical process; or the study described in detail any process related to the pre-analytical handling of the samples that could affect the results, and a comparison of the results according to the different procedures was supplied; "high risk of bias": if this information was not reported.

5. Description of metabolite extraction methods and analytical techniques – "low risk of bias": if the study reported in detail the metabolite extraction methods and analytical techniques used; "high risk of bias": if this information was not reported.

6. Blinded interpretation of metabolomic results to fetal growth status of participants – "low risk of bias": if metabolomic results were interpreted blinded to fetal growth status of participants; "high risk of bias": if metabolomic results were not interpreted blinded to fetal growth status of participants.

7. Control for potential confounding variables – "low risk of bias": the main potential confounding variables were identified and accounted for in the design and analysis; "high risk of bias": the main potential confounding variables were not identified and/or accounted for in the design and analysis.

8. Avoidance of overfitting in statistical models – "low risk of bias": if the models were validated in an independent set of samples or used some approach to deal with overfitting; "high risk of bias": if the models were not validated in an independent set of samples or did not use some approach to deal with overfitting; or if the study used the same sample for the training and validation sets.

If there was insufficient information available to make a judgement about these items, then they were scored as "unclear risk of bias".

### Data extraction
Data were extracted from each included study using a specially designed form for capturing information on authors, publication date, study characteristics (experimental design, setting, follow-up period, attrition and exclusions from the analysis, prospective or retrospective data collection, blinded interpretation of metabolomic results), participants (selection, inclusion and exclusion criteria, case definition, control definition, number of women/neonates in each study group, baseline characteristics, and country and date of recruitment), biological samples (sampling time, sample collection and storage, frequency of sampling, handling of specimens, pre-analytical procedures, metabolite extraction methods, and analytical techniques), metabolomics data analysis (feature extraction, compound identification, statistical analysis and interpretation), and metabolites (reported metabolite identity by the authors of the paper (ID), and metabolites with statistically significant differences in concentration between the FGR/SGA and reference/control groups).

### Data synthesis
Substantial heterogeneity in the analytical platforms used, variation in multivariate analyses, and incomplete and heterogeneous reporting of metabolite data and summary statistics prevented us from performing a quantitative meta-analysis and precluded us from determining the average fold changes of metabolite levels across all studies for any metabolite.

We separately analysed metabolite alterations as reported in the publications, in 10 types of biological sample: maternal plasma or serum, urine, faeces and hair (collected at ≤20 or >20 weeks' gestation, or in the peripartum period), amniotic fluid, placenta, umbilical cord blood, neonatal blood and urine, and breast milk. Metabolites were identified according to the reported identity (ID) or common name with a subsequent standardisation conducted by using the Human Metabolomic Database (HMDB) to assign unique identifiers thereby avoiding synonymous names. Given most included studies reported the directionality of the identified metabolites, we initially selected and counted the total number of significantly up- and down-regulated metabolites, as compared with the corresponding reference group, in each study. A metabolite was considered as statistically significantly up- or down-regulated, as reported by the authors of individual studies, regardless of the p value threshold used for defining statistical significance and the use of tests for correcting multiple comparisons. Eighty-one percent of the included studies used a $p$ value < 0.05 to determine significance.

Then, we summarised the significantly up- and down-regulated metabolites that were reported in at least more than one study, grouping them into maternal samples collected at ≤20 or >20 weeks' gestation/peripartum period, and neonatal samples (all differentiated by the type of biological sample), placental samples, and umbilical cord blood samples, according to the metabolite's direction of change between studies. The trend of one metabolite was considered "consistent" if it showed the same direction of change in all studies within the same parameter and biological sample (e.g. up- or down-regulated in all studies reporting maternal plasma). Otherwise, the metabolite's trend was considered "inconsistent".

Finally, we imported the significantly up- or down-regulated metabolites in at least more than one study with a consistent trend to MetaboAnalyst 5.0 online software (https://www.metaboanalyst.ca)[85] for separate pathway analysis in each of the aforementioned groups of biological samples. The software allows the most relevant pathways involved in the conditions under study to be identified. A metabolic pathway was considered to be significantly enriched if its adjusted $p$ value (false discovery rate, FDR) was <0.05. Since we were testing many pathways at the same time, the statistical p values from enrichment analysis were further adjusted for multiple testing (Holm p, $p$ value adjusted by Holm–Bonferroni method; and FDR p, $p$ value adjusted using false discovery rate). The impact value (from 0.0 to 1.0) represents the relative importance of the pathway: the higher the impact value, the more relevant is the pathway in the condition under study.

### Reporting summary
Further information on research design is available in the Nature Portfolio Reporting Summary linked to this article.

## Data availability
The findings from this study are supported by data extracted from published literature. The relevant studies were identified through a systematic literature review and can all be accessed online as referenced in the current paper[27–74]. Study characteristics of all relevant studies included in the analyses are also provided in Table 1. All data supporting the findings described in this manuscript are available in the article and in the Supplementary Information and from the corresponding authors upon request. Data that support the findings of this study have been deposited in the Mendeley Data Repository: https://data.mendeley.com/datasets/4rsm8zv5x3/1 (https://doi.org/10.17632/4rsm8zv5x3.1). Source data are provided with this paper.

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

## Acknowledgements

This research was supported by a grant (INV029003) from the Bill & Melinda Gates Foundation. The funder had no role in the design or conduct of the study; collection, management, analysis, or interpretation of the data; preparation, review or approval of the manuscript or the decision to submit the manuscript for publication.

## Author contributions

A.C-A., J.V., and M.R. conceived and designed the study. A.C-A. and M.R. performed the literature search. A.C-A. and J.V. selected the studies, assessed their risk of bias, and extracted data. A.C-A. performed the data analysis. A.C-A. and J.V. wrote the initial draft. All authors (A.C-A., J.V., M.R., A.T.P., L.D.R., and S.H.K.) contributed to the data interpretation, revised the manuscript for important intellectual content, and approved the final version of the manuscript for publication. J.V. and S.H.K. acquired funding.

## Competing interests

A.T.P. is supported by the Oxford Partnership Comprehensive Biomedical Research Centre with funding from the NIHR Biomedical Research Centre (BRC) funding scheme. The views expressed herein are those of the authors and not necessarily those of the NHS, the NIHR, the Department of Health or any of the other funders. All other authors declare no competing interests.
