## [Peer Review file · Nature Communications]

Metabolomic signatures associated with fetal growth restriction and small for gestational age: a systematic review

Corresponding Author: Professor Jose Villar

This manuscript has been previously reviewed at another journal. This document only contains reviewer comments, rebuttal and decision letters for versions considered at Nature Communications.

Version 0:

Reviewer comments:

Reviewer #1

(Remarks to the Author)

The authors have broadly addressed major concerns.

Reviewer #2

(Remarks to the Author)

I reviewed this paper when it was initially submitted. Thank you for the opportunity to review the revised version. I would like to congratulate the authors on their efforts to address many of the helpful comments provided by the previous review. Most have been addressed.

My key message previously was the need for simplification as the area is broad and quite a lot of results are presented. I feel this has been addressed now. This is a paper dense with results - it is however easier to comprehend the multitude of findings. The discussion is also now more comprehensive, and the structure will be of benefit to the readership. I have not identified any further significant concerns.

Reviewer #3

(Remarks to the Author)

In the manuscript "Metabolomic signatures associated with fetal growth restriction and small for gestational age: a systematic review", Conde-Agudelo et al aimed to understand the pathophysiology of fetal growth restriction (FGR) and small for gestational age (SGA) by systematically searching, evaluating and summarizing the evidence of metabolites and associated pathways in both maternal and fetal/neonatal tissues. The authors were able to select a profound amount of data with 56 metabolites investigated in more than one study. However, whether specific mechanisms are causing FGR or SGA or are only correlated remains to be elucidated.

The amount of evidence collected by the authors is impressive and a promising base for this study.

After thorough reading this interesting manuscript, I have following questions and comments for the authors:

The manuscript is rather long and would benefit from shortening (word limit for Nature Communications articles are 5000 words according to the author's guideline). In my comments, I suggest various parts that could be summarized.

Introduction:

1. The authors describe well that FGR and SGA are defined differently in the introduction. I miss this differentiation later in the manuscript, e.g. when describing the results or in the discussion.
2. Screening for FGR is a prominent aspect discussed in the introduction. This might could lead the reader to think that finding biomarkers for screening for FGR is an aim of this study. The text could benefit from restructuring the paragraph beginning line 92 by focusing more on the unknown pathophysiology.

Results:

1. According to the comments from the previous reviewing process, the result section is quite hard to read. For example, the text uses many parentheses disturbing the reader's flow. Since the data is summarized in the tables, less information in the text is needed when referring to the corresponding table.
2. Table 1 would benefit from restructuring: The column "Case and control definition (sample size)" should be divided into two columns Cases (n) and Control (n) highlighting the heterogeneous definitions. A clear description and highlighting the definitions of SGA and FGR is crucial. Adding a column stating the gestational age at birth in each study would be important since this directly affects the definition of SGA and FGR. Another column adding the information of the origin of the tissue (maternal or fetal/neonatal) would be helpful.
3. Why is the study Horgan et al (2010) included? The study utilized an in-vitro approach of placental villous explants which would meet the exclusion criteria.
4. The study characteristics are well described, but the amount of details can be confusing. The authors should try to highlight the main findings in this section and refer for details to Table 1. For example, in line 128 and ongoing, studies collecting multiple biological samples are described in detail. By referring to Table 1, this could be shortened to one sentence: Fourteen studies collected multiple biological samples (Table 1).
5. How many non-significant metabolites have been analyzed? It would be helpful to at least add the number to avoid selective reporting bias.
6. Information (paragraph starting line 177) about statistical multivariate approaches is not relevant for the outcome of the systematic review. Why is it reported?
7. The Results section could benefit from adding the subheadings maternal tissues, newborn tissues and fetal tissue.
8. The Results would benefit from more consistency in presenting the data. E.g. the findings of single studies reporting on significant differences are highlighted in detail in the text in some sections e.g. for maternal plasma at < 20 weeks' gestation or placenta. This is not the case in umbilical cord blood. Considering the systematic approach, the reporting in the results should be consistent and similar structured for all evidence. I would suggest to not highlight results of single studies considering that the strength of systematic reviews is summarizing the evidence.
9. Have the authors compared the case and control populations of the different studies summarized in the analysis of metabolic pathways? It would be important to note if the definition of FGR or SGA in these studies are comparable before connecting the data.

Discussion:

1. The Discussion section would benefit from more subheadings.
2. According to the comments from the previous reviewing process, the text would benefit from more concise descriptions of the main findings.
3. The paragraph about the risk of bias assessment (starting line 515) is repeating the data given in the results. It should be summarized and focusing on the main message.
4. As main outcome, the authors wanted to investigate the pathophysiology of FGR and SGA. Could the authors highlight their findings for this outcome more? I miss the focus that was intended by the introduction.
5. While describing the strength of the study (line 476), bullet points (2), (3) and (5) are required standards of systematic reviews and are indicated by bullet point (1). There is no need to describe them.
6. The authors describe the heterogeneous definitions of FGR and SGA as a limitation of the study. This should be highlighted more since it makes the included studies not comparable.
7. Could the authors describe the term "clinical intervention metabolomic studies" (line 555-556)?

Methods:

1. In the registered PROSPERO protocol, preterm birth has been described as an outcome. Why is it not reported in the present study? To avoid selection bias, it is imperative to report preterm birth and include in the discussion.
2. It is standard in a systematic review to first describe the search strategy and then inclusion and exclusion criteria (Eligibility criteria).
3. When has the search been performed? Has it been updated? Please state this in the text.
4. Has the search included a time-limit? If yes, the authors need to explain the reasoning.
5. Could the authors provide the keywords in MeSH terms and an example of the search strategy for one of the databases to ensure reproducibility?
6. How have the authors conducted the search in other sources? (Fig. 1) Why have none been found?
8. The description of the modified QUADOMICS tool is very detailed and could be summarized.

Version 1:

Reviewer comments:

Reviewer #3

(Remarks to the Author)

I thank the authors to revise my raised points thoroughly. All major points have been addressed. The revised version of the manuscript is more understandable and summarizes the collected evidence better. The reporting of the methodology has significantly improved.

I would like to address one minor concern: I suggest the authors to double-check the references and citations in the revised manuscript.

For example, in line 136, the wrong references have been cited. There should only be one study referred to instead of 22. Similar in line 137 and 140. And in line 223, the authors mention five studies, but only four studies are cited.

REVIEWER 1

REVIEWER 1, POINT 1

Remarks to the Author:

The authors have broadly addressed major concerns.

Reply

We thank the Reviewer for this kind remark.

REVIEWER 2

REVIEWER 2, POINT 1

Remarks to the Author:

I reviewed this paper when it was initially submitted. Thank you for the opportunity to review the revised version.

I would like to congratulate the authors on their efforts to address many of the helpful comments provided by the previous review. Most have been addressed.

My key message previously was the need for simplification as the area is broad and quite a lot of results are presented. I feel this has been addressed now. This is a paper dense with results - it is however easier to comprehend the multitude of findings. The discussion is also now more comprehensive, and the structure will be of benefit to the readership.

I have not identified any further significant concerns.

Reply

We thank the Reviewer for these kind remarks.

REVIEWER 3

Remarks to the Author:

In the manuscript "Metabolomic signatures associated with fetal growth restriction and small for gestational age: a systematic review", Conde-Agudelo et al aimed to understand the pathophysiology of fetal growth restriction (FGR) and small for gestational age (SGA) by systematically searching, evaluating and summarizing the evidence of metabolites and associated pathways in both maternal and fetal/neonatal tissues. The authors were able to select a profound amount of data with 56 metabolites investigated in more than one study. However, whether specific mechanisms are causing FGR or SGA or are only correlated remains to be elucidated.

The amount of evidence collected by the authors is impressive and a promising base for this study. After thorough reading this interesting manuscript, I have following questions and comments for the authors:

The manuscript is rather long and would benefit from shortening (word limit for Nature Communications articles are 5000 words according to the author's guideline). In my comments, I suggest various parts that could be summarized.

REVIEWER 3, POINT 1

Introduction

1. The authors describe well that FGR and SGA are defined differently in the introduction. I miss this differentiation later in the manuscript, e.g. when describing the results or in the discussion.

Reply

A major problem in studying FGR is that there is no gold standard for identifying the condition, as the genetically determined growth potential of a fetus is unknown. A range of different proxies have been described. As none of these is a perfect measure of the condition, distinguishing FGR pregnancies from those that are not FGR using any of the available methods is inevitably associated with misclassification, i.e. in every analysis of a cohort of pregnancies, there will be cases defined as FGR where the baby was healthy and cases where the baby had FGR but was wrongly classified as normal. On the other hand, a baby might have a birthweight within the 'normal' range (10-90th centile) but have been affected by FGR as the true, and unknown, genetically determined centile was the 95th.

Several authors use the terms SGA and FGR interchangeably, although the majority of SGA babies are healthy and constitutionally small. Therefore, it is foreseeable that most included studies in our systematic review included both FGR and SGA babies. This was the main reason why the objective of our study was to identify metabolomic signatures in maternal and newborn tissues and body fluids samples to understand the pathophysiology of FGR/SGA.

In keeping with the same Reviewer's suggestion in Point 17, in the new "Strengths and limitations" subsection of the Discussion section, we have replaced the statement "Second, a major source of variation across studies was the use of very different definitions of FGR/SGA..." with "Second, a major source of variation across studies was: 1) the use of very different definitions of FGR and SGA (that are often wrongly used interchangeably in the literature) and categories of reference groups, and 2) the failure to recognise the syndromic nature of these two anthropometric and clinical entities that have multiple inter-related aetiologies and risk factors. These limitations considerably undermine both the internal and external validity of studies. Hence, it is possible..."

REVIEWER 3, POINT 2

2. Screening for FGR is a prominent aspect discussed in the introduction. This might lead the reader to think that finding biomarkers for screening for FGR is an aim of this study. The text could benefit from restructuring the paragraph beginning line 92 by focusing more on the unknown pathophysiology.

Reply

We mentioned in a short statement (lines 92-94) that accurate biomarkers are needed not only for screening but also for developing preventive and treatment strategies for FGR. Therefore, we do not think that, after reading this short statement, readers could think that the objective of this systematic review was to find biomarkers for FGR screening. Instead, we clearly state that the aim of our systematic review was to identify metabolomic signatures in tissues and biofluids of pregnant women, placentas, umbilical cords and newborns associated with FGR/SGA compared to the corresponding reference group. We think it is not necessary to restructure the paragraph beginning line 92. Thus, the Reviewer's comment has not resulted in a change in the revised version of the manuscript.

REVIEWER 3, POINT 3

Results

1. According to the comments from the previous reviewing process, the result section is quite hard to read. For example, the text uses many parentheses disturbing the reader's flow. Since the data is summarized in the tables, less information in the text is needed when referring to the corresponding table.

Reply

We used parentheses to include summary measures such as total number of studies, medians, and percentages, which are not reported in the tables. However, we agree with the Reviewer's comment that less information in the text is needed when referring to the corresponding table, mainly Table 1.

Thus, the Reviewer's comment has resulted in a change in the revised version of the manuscript: in several subsections of the Results section reporting on metabolomic profiles, we have shortened the word count from 1956 to 1631, without affecting the content.

REVIEWER 3, POINT 4

2. Table 1 would benefit from restructuring: The column "Case and control definition (sample size)" should be divided into two columns Cases (n) and Control (n) highlighting the heterogeneous definitions. A clear description and highlighting the definitions of SGA and FGR is crucial. Adding a column stating the gestational age at birth in each study would be important since this directly affects the definition of SGA and FGR. Another column adding the information of the origin of the tissue (maternal or fetal/neonatal) would be helpful.

Reply

We thank the Reviewer for this helpful suggestion. In Table 1, we have divided the column "Case and control definition (sample size)" into two columns: "Case definition (n)" and "Control definition (n)", and added a column giving the gestational age at birth in each included study. Finally, as we included, in the original table, the column

“Biological sample; sampling time”, which indicates the origin of the tissue, we have not added a column indicating the origin of the tissue.

REVIEWER 3, POINT 5

3. *Why is the study Horgan et al (2010) included? The study utilized an in-vitro approach of placental villous explants which would meet the exclusion criteria.*

Reply

The study by Horgan et al (2010) assessed changes in the metabolic footprint of placental explant-conditioned medium cultured in different oxygen tensions from placentas of SGA and normal pregnancies. Therefore, although this study utilised an *in vitro* approach, it met the inclusion criteria (observational studies that reported on associations between metabolites measured using any metabolomic technology in tissues and biofluids of women with a singleton pregnancy or singleton newborns and FGR or SGA diagnosed by criteria defined by the authors).

The Reviewer's comment has resulted in a change in the revised version of the manuscript. In the subsection "Eligibility criteria" (page 19), we have deleted the words "and *in vitro*". We thank the Reviewer for bringing this error to our attention.

REVIEWER 3, POINT 6

4. *The study characteristics are well described, but the amount of details can be confusing. The authors should try to highlight the main findings in this section and refer for details to Table 1. For example, in line 128 and ongoing, studies collecting multiple biological samples are described in detail. By referring to Table 1, this could be shortened to one sentence: Fourteen studies collected multiple biological samples (Table 1).*

Reply

We agree with the Reviewer's suggestion of shortening the sentence beginning line 128. Their comment has resulted in a change in the revised version of the manuscript. In the "Selection, characteristics and risk of bias of studies" subsection of the revised manuscript, we have replaced the statement "Fourteen studies collected multiple biological samples: 11 studies collected samples from both maternal plasma or serum and umbilical cord blood,^{29,31,32,40,42,43,54,56,64,73,74} one from maternal plasma and maternal urine,⁵³ one from maternal serum and maternal faeces,⁷⁰ and one from maternal serum, placenta, and umbilical cord blood.⁷²" by "Fourteen studies^{29,31,32,40,42,43,53,54,56,64,70,72-74} collected multiple biological samples (Table 1)".

In addition, in the penultimate paragraph of the same subsection, we have deleted the statement "because one, reported there were no significant differences in metabolite concentrations between the SGA and AGA groups in umbilical cord blood, and the other found several altered metabolites in urine of growth-restricted newborns, which were not reported in either of the two other studies that assessed metabolomic profiles in newborn urine."

REVIEWER 3, POINT 7

5. How many non-significant metabolites have been analyzed? It would be helpful to at least add the number to avoid selective reporting bias.

Reply

Among the 48 included studies, 13 did not report the number of non-significant metabolites between the study groups, 10 reported it unclearly, and only 25 (52%) clearly reported it. Overall, the total number of non-significant metabolites reported in these 25 studies was 6198. However, we think that including the number of non-significant metabolites from only 25 studies in the Results section would represent a selective reporting bias. Therefore, we prefer not to include the total number of non-significant metabolites.

REVIEWER 3, POINT 8

6. Information (paragraph starting line 177) about statistical multivariate approaches is not relevant for the outcome of the systematic review. Why is it reported?.

Reply

Multivariate analysis methods aim to differentiate between classes in highly complex datasets, despite within class variability. There are a variety of multivariate analysis methods all of which have their own strengths and weaknesses. In designing experiments for metabolomics studies, the choice of multivariate analysis method must be driven by the data and the experimental goals. For example, in exploratory studies where metabolomic differences between experimental groups may be unknown or unpredictable, initial application of principal component analysis (PCA) provides an informative first look at the dataset structure and relationships between groups. Then, the results of PCA analyses can be used to formulate an initial biological conclusion, which partial least square (PLS) or orthogonal PLS can then verify and test in more detail. Therefore, we think it is important readers know the statistical multivariate approaches that were used in the included studies and we have not altered the manuscript as a result of the Reviewer's comment.

REVIEWER 3, POINT 9

7. The Results section could benefit from adding the subheadings maternal tissues, newborn tissues and fetal tissue.

Reply

In the Results section of the original manuscript, we included the subheadings "Metabolomic profiles" in maternal plasma or serum, maternal hair, maternal urine, maternal faeces, amniotic fluid, placenta, umbilical cord blood, newborn dried blood spots, newborn urine, and breast milk. Therefore, we do not think it is necessary to add the subheadings suggested by the Reviewer.

REVIEWER 3, POINT 10

8. The Results would benefit from more consistency in presenting the data. E.g. the findings of single studies reporting on significant differences are highlighted in detail in the text in some sections e.g. for maternal plasma at < 20 weeks' gestation or placenta. This is not the case in umbilical cord blood. Considering the systematic approach, the reporting in the results should be consistent and similar structured for all evidence. I would suggest to not highlight results of single studies considering that the strength of systematic reviews is summarizing the evidence.

Reply

The identification of altered metabolites and metabolic pathways that are associated with FGR and/or SGA in plasma or serum of pregnant women at ≤ 20 weeks' gestation is crucial for the discovery of novel biomarkers for the potential prediction of these conditions and for determining target metabolic pathways for prevention and treatment. This explains why we highlighted the findings of single studies reporting on metabolomic profiles in maternal plasma or serum at ≤ 20 weeks' gestation.

However, we agree with the Reviewer's suggestion not to highlight the results of single studies. Thus, the Reviewer's comment has resulted in a change in the revised version of the manuscript. In several subsections of the Results section reporting on metabolomic profiles, we have summarised the results of all studies instead of highlighting results in each single study, as suggested. We think the reporting of the results is now consistent and similarly structured for all evidence.

REVIEWER 3, POINT 11

9. Have the authors compared the case and control populations of the different studies summarized in the analysis of metabolic pathways? It would be important to note if the definition of FGR or SGA in these studies are comparable before connecting the data.

Reply

This issue has partially addressed in the reply to Reviewer 3's Point 1. When performing the analysis of metabolic pathways, we did not compare the case and control populations of the different studies because it is very likely that most included studies, regardless of the definition of FGR or SGA used, included both FGR and SGA fetuses/newborns. It was the main reason why the objective of our study was to identify metabolomic signatures in maternal and newborn tissues and body fluids samples to understand the pathophysiology of FGR/SGA without distinguishing between these conditions. The use of different definitions of FGR/SGA and categories of reference groups, and the lack of recognition of the syndromic nature of these two anthropometric and clinical entities, which undermine both the internal and external validity of studies, were highlighted as a limitation of our study in the Discussion section of the original submitted manuscript, which has now been strengthened in keeping with the same Reviewer's suggestion in Point 17.

REVIEWER 3, POINT 12

Discussion

1. *The Discussion section would benefit from more subheadings.*

Reply

We thank the Reviewer for this helpful comment, which has resulted in a change in the revised manuscript. In the Discussion section, we have added the following subheadings: “Principal findings”, “Comparison with existing literature”, and “Strengths and limitations”.

REVIEWER 3, POINT 13

2. *According to the comments from the previous reviewing process, the text would benefit from more concise descriptions of the main findings.*

Reply

We agree with the Reviewer’s comment, which has resulted in a change in the revised manuscript. In the new “Principal findings” subsection of the Discussion section, we have deleted the second paragraph (“Unfortunately, only one study assessed metabolomic profiles in both maternal samples at ≤ 20 weeks’ gestation and the corresponding umbilical cord blood samples.....There were no significant associations between umbilical cord serum concentrations of amino acids, phosphatidylcholines, and acylcarnitine and SGA”) making the description of the main findings more concise, as suggested.

REVIEWER 3, POINT 14

3. *The paragraph about the risk of bias assessment (starting line 515) is repeating the data given in the results. It should be summarized and focusing on the main message.*

Reply

We agree with the Reviewer’s comment, which has resulted in a change in the revised version of the manuscript. In the new “Strengths and limitations” subsection of the Discussion section, we have replaced the statement “In our review, only 27% of included studies met at least six of the eight criteria for low risk of bias; only 23% clearly reported that the models were validated in an independent set of samples or some approach had been used to deal with overfitting; and only 29% did not take into account potential confounding factors in the analyses. In addition, most included studies” with “In our review, only just over a quarter of included studies met at least six of the eight criteria for low risk of bias. In addition, most included studies”

REVIEWER 3, POINT 15

4. As main outcome, the authors wanted to investigate the pathophysiology of FGR and SGA. Could the authors highlight their findings for this outcome more? I miss the focus that was intended by the introduction.

Reply

The aim of our systematic review was specifically to identify metabolomic signatures associated with FGR/SGA in maternal and newborn tissues and body fluids samples in order to shed new light on the metabolic pathways involved in the pathophysiology of these conditions. We did not intend to investigate the pathophysiology of FGR and SGA, as the Reviewer has suggested. Hence, their comment has not resulted in a change in the revised version of the manuscript.

REVIEWER 3, POINT 16

5. While describing the strength of the study (line 476), bullet points (2), (3) and (5) are required standards of systematic reviews and are indicated by bullet point (1). There is no need to describe them.

Reply

We agree with the Reviewer's comment that bullet points (3) and (5) are required standards of systematic reviews and there is no need to describe them. However, the inclusion of the largest number of mostly recent studies reported from different populations throughout the world, bullet point (2), is a major strength of our systematic review, which we believe should be highlighted. Hence, we have limited the change to the revised manuscript in the new "Strengths and limitations" subsection of the Discussion section by only deleting bullet points (3) and (5).

REVIEWER 3, POINT 17

6. The authors describe the heterogenous definitions of FGR and SGA as a limitation of the study. This should be highlighted more since it makes the included studies not comparable.

Reply

We thank the Reviewer for this helpful suggestion, which has also been addressed in the responses to points 1 and 11 of Reviewer 3. In the new "Strengths and limitations" subsection of the Discussion section, we have replaced the statement "Second, a major source of variation across studies was the use of very different definitions of FGR/SGA..." with "Second, a major source of variation across studies was: 1) the use of very different definitions of FGR and SGA (that are often wrongly used interchangeably in the literature) and categories of reference groups, and 2) the failure to recognise the syndromic nature of these two anthropometric and clinical entities that have multiple inter-related aetiologies and risk factors. These limitations considerably undermine both the internal and external validity of studies. Hence, it is possible..."

REVIEWER 3, POINT 18

7. Could the authors describe the term “clinical intervention metabolomic studies” (line 555-556)?

Reply

The term “clinical intervention metabolomic studies” refers to studies in which metabolomic signatures could guide the classification of diseases and stratification of patients, provide biomarkers of drug response, safety, and interactions (pharmacometabolomics), and monitor the progression or improvement of diseases in clinical trials. We think the term is self-explanatory. The comment of the Reviewer has not resulted in a change in the revised version of the manuscript.

REVIEWER 3, POINT 19

Methods:

1. In the registered PROSPERO protocol, preterm birth has been described as an outcome. Why is it not reported in the present study? To avoid selection bias, it is imperative to report preterm birth and include in the discussion.

Reply

The metabolomic signatures in tissues and biofluids of pregnant women, placentas, umbilical cords and newborns associated with preterm birth phenotypes will be reported in a separate paper.

REVIEWER 3, POINT 20

2. It is standard in a systematic review to first describe the search strategy and then inclusion and exclusion criteria (Eligibility criteria).

Reply

We agree with the Reviewer’s comment, which has resulted in a change in the revised manuscript. In the Methods section of the revised manuscript, we have first described the literature search strategy and then the eligibility criteria, as suggested.

REVIEWER 3, POINT 21

3. When has the search been performed? Has it been updated? Please state this in the text.

Reply

The initial search was performed from 1 June 2023 to 15 June 2023. Searches were re-run on a monthly basis until 2 January 2024. The Reviewer’s comment has resulted in a change in the revised version of the manuscript. In the “Literature search”

subsection of the revised manuscript, we have added the following statement: “The initial search was performed from 1 June 2023 to 15 June 2023. Searches were re-run on a monthly basis until 2 January 2024”.

REVIEWER 3, POINT 22

4. Has the search included a time-limit? If yes, the authors need to explain the reasoning.

Reply

We thank the Reviewer for this helpful suggestion. We searched databases for studies published from 1998, the year that the term metabolomics was introduced, to 31 December 2023. Thus, in the “Literature search” subsection of the revised manuscript, we have replaced the statement “We searched MEDLINE, EMBASE, LILACS, CINAHL, Scopus, Web of Science, and the Cochrane Central Register of Controlled Trials (all from 1998 to December 31, 2023) using a combination of” by “We searched MEDLINE, EMBASE, LILACS, CINAHL, Scopus, Web of Science, and the Cochrane Central Register of Controlled Trials (all from 1998, the year that the term metabolomics was introduced, to 31 December 2023) using a combination of”.

REVIEWER 3, POINT 23

5. Could the authors provide the keywords in MeSH terms and an example of the search strategy for one of the databases to ensure reproducibility?

Reply

(1) Keywords in MeSH terms:

Metabolomics
Metabolome
Lipidomics
Oxylipins
Proton Magnetic Resonance Spectroscopy
Chromatography, Liquid
Chromatography, Gas
Chromatography, High Pressure Liquid
Fetal Growth Retardation
Infant, Small for Gestational Age

(2) Search strategy in PubMed

((((((((((((((((metabolomic*) OR (metabonomic*) OR (metabolome*) OR (metabolite*)) OR (lipidomic*) OR (oxylipins)) OR (proton nuclear magnetic resonance)) OR (proton magnetic resonance spectroscopy)) OR (liquid chromatograph*)) OR (gas chromatograph*)) OR (high-performance liquid chromatograph*)) OR (HPLC)) OR (ultra-performance liquid chromatograph*)) OR (UPLC))) AND (((((((((fetal growth restriction) OR (fetal growth retardation)) OR (intrauterine growth restriction)) OR (intrauterine growth retardation)) OR (impaired fetal growth)) OR (small for gestational age)) OR (small for date)) OR (small for gestation)))

We leave the inclusion of this search strategy as online Supplementary information to the discretion of the editor. The comment of the Reviewer has not resulted in a change in the revised version of the manuscript.

REVIEWER 3, POINT 24

6. How have the authors conducted the search in other sources? (Fig. 1) Why have none been found?

Reply

The search in other sources was conducted in Google Scholar, proceedings of congresses and scientific meetings on obstetrics, maternal-fetal medicine and omics technologies, reference lists of identified studies, previously published systematic reviews, and review articles. We did not identify additional citations in these sources. This was probably due to the previous comprehensive literature search we performed in seven electronic databases.

REVIEWER 3, POINT 25

8. The description of the modified QUADOMICS tool is very detailed and could be summarized.

Reply

We think readers should know in detail the modified QUADOMICS tool we used for assessing the risk of bias in the included studies. However, we leave the inclusion of the complete description of the modified QUADOMICS tool as online Supplementary information to the Editor's discretion.

Please, see below the point-by-point response to the Reviewers' comments.

Sincerely yours,

Agustin Conde-Agudelo and Jose Villar, on behalf of all of the co-authors

REVIEWER 3

REVIEWER 3, POINT 1

I thank the authors to revise my raised points thoroughly. All major points have been addressed. The revised version of the manuscript is more understandable and summarizes the collected evidence better. The reporting of the methodology has significantly improved.

Reply

We thank the Reviewer for these kind remarks.

REVIEWER 3, POINT 2

I would like to address one minor concern: I suggest the authors to double-check the references and citations in the revised manuscript. For example, in line 136, the wrong references have been cited. There should only be one study referred to instead of 22. Similar in line 137 and 140. And in line 223, the authors mention five studies, but only four studies are cited.

Reply

We thank the Reviewer for bringing this mistake to our attention. Errors in the citation of these references were due to typographical errors. The Reviewer's comment has resulted in a change in the revised manuscript. We have double-checked the references and citations and have corrected the errors detected by the Reviewer in lines 101-110 of the revised manuscript. No other errors in citing references were detected in the manuscript.